

# The Pareto effect in tipping social networks: from minority to majority

Jordan Everall[1], Jonathan. F Donges[23], Ilona. M. Otto[12]

[1]Wegener Center for Climate and Global Change, University of Graz, Austria
[2]Potsdam Institute for Climate Impact Research (PIK), Member of the Leibniz Association, Potsdam, Germany
[3]Stockholm Resilience Centre, Stockholm University, Stockholm, Sweden

*Correspondence to*: Jordan Everall (jordan.everall@uni-graz.at)

**Abstract.** How do social networks tip? A popular theory is that a small minority can affect network, or population wide change. This effect is roughly consistent with the properties of the Pareto principle, a semi-quantitative law which suggests that in many systems, 80% of effects are produced by only 20% of the causes. In the context of the transition to net-zero emissions, this vital 20% can be a critical instigator of social tipping, a process which can rapidly accelerate social norm change. In this work, we ask whether the Pareto effect can be observed in social systems by conducting a literature review with a focus on social norm diffusion and complex contagion on social networks. By collecting simulation and empirical results of social tipping events over a wide disciplinary, and parametric space, we are able to see the existence of shared behaviour across studies. Based on a compiled dataset, we show general support for the existence of a tipping point which occurs at around 25% of the total population in susceptible social systems. Around this critical mass, there is a high likelihood of a social tipping event, where a large minority is then quickly "tipped". Additionally, we were able to show a range of critical masses where social tipping is possible, these values lie roughly between 10% and 45%. Finally, we also provide practical advice for facilitating norm changes under uncertainty, difficult social norm transitions, and social groups resistant to change.

## 1 Introduction

Nonlinear dynamics, under which social tipping processes can be considered, have been covered comprehensively by both natural (Strogatz, 2019) and social scientists over the last century. Famous examples are Granovetter (1973), who showed that in certain social network structures a select minority can alter macro scale information flow, and Schelling (1971), who demonstrated that a small individual racial preference in a minority, can lead to completely segregated neighbourhoods. Some contemporary examples focus on rapid shifts in smoking behaviour (Nyborg et al., 2016) and "critical mass phenomenon", whereby a minority participation (25-30%) in a collective event can engage a remaining majority (Andreoni



et al., 2021; Centola et al., 2018). As recognition of the close coupling of social and physical systems characteristic of the Anthropocene has mounted (Steffen et al., 2018; Lenton, 2020), so too has research on social tipping processes in the context of climate, and global environmental change due to their potential as mechanisms of rapid societal transformation (David Tàbara et al., 2018; Nyborg et al., 2016; Otto et al., 2020b; Lenton, 2020; Westley et al., 2011). Here, this new layer of

tipping-scholarship is centred around deliberately bringing about social change through targeted action on tipping elements "during sensitive intervention points" (Farmer et al., 2019), or moments of opportunity that trigger a tipping point. It must be noted at this point, that the definition of tipping points in a Socio-Ecological Systems (SES) context is not uniform. We seek to provide a concise summary and provide a guide for understanding these concepts in the context of this work in section 2.1.

New research in this sector can be broken down into analyses and analytical frameworks. Key examples of the first are seen in Otto et al., (2020) who identifies several concrete societal tipping elements and timescales through expert elicitation, likewise Farmer et al., (2019), and Lenton (2020) also indicate critical points for intervention in financial, energy, resource and governance systems, to name a few. Frameworks focus more generally on processes, phases and conceptualisation of "radical" socioecological transitions (Feola, 2015). More recent work (Winkelmann et al., 2020), proposes a framework

including a more detailed understanding of social tipping mechanisms. In which, critical elements; such as social network properties (e.g. polarisation, clustering, and modularity), agency, temporospatial scales, and dynamics; such as social contagion and adaptation of networks are explicitly included. Much of this work focuses strongly on the existence or identification of social tipping points, the need to trigger them, and their value in the sustainability transition. There is an abundance of theory specific to modelling social tipping in social-environmental-systems, as opposed to general social

systems (Lade et al., 2017; Schwarz and Ernst, 2009; Schwarz et al., 2020; Müller-Hansen et al., 2017), and a body (Schleussner et al., 2016; Schunck et al., 2021; Geier et al., 2019; Andersson et al., 2020; Frei et al., 2023) of recent empirical work in statistical physics, network science, and computational social science also acknowledging their applications to SES transformation.

A theme openly discussed in the literature is the prediction of social tipping points, and whether this is at all possible at large

scales, in complex social-ecological-systems (Bentley et al., 2014). Closely related is the question of the general existence of social tipping points, implying fundamentally universal and scale-free social dynamics related to tipping processes. It is largely understood that due to complexity, heterogeneity, and dependence on context, any general tipping point is difficult to predict (Winkelmann et al., 2020; Bentley et al., 2014). In some circumstances they may not even exist (Ferraz de Arruda et al., 2023). Despite this, there seems to be evidence for tipping, or as conceptualised in network theory (Guilbeault et al.,

2018); contagion dynamics, across and between societies, scopes, and organisms (Dodds and Watts, 2004)). A not insignificant number of overlaps or co-occurrences between empirical and modelling results of social contagion processes from various disciplines confirm this  (Centola et al., 2018; Andreoni et al., 2021; Xie et al., 2011; Wiedermann et al., 2020)(Centola et al., 2018, Andreoni et al., 2021,  Xie et al., 2011, Wiedermann et al., 2020) . While it is highly unlikely the





employed methods will ever quantitatively predict tipping points across systems, their results can be used to inform a range
of scenarios under which tipping is more likely.

Tackling these questions surrounding social tipping points requires the study of social networks. Social networks are critical
in understanding and studying social tipping processes (Granovetter, 1978; Watts and Dodds, 2007; Watts and Strogatz,
1998). Beginning here, we focus on quantifying universal or general trends in social tipping literature across several
disciplines. Undertaking such a task can be prohibitively difficult due to the high dimensionality of a data set when compiled
across disciplines, inconsistency of terms, and the complexity of social tipping in general where there are many confounders
present (Milkoreit, 2023). This is made especially difficult when the intent is to include a quantitative analysis, where
variables such as critical mass, tipping thresholds (macroscopic and individual) etc. cannot be mapped one to one. In order to
ensure robust results, we take a conservative approach to any data reported and focus on marginal effects of individual
factors where many explanatory variables are involved. We also for example provide a range of social tipping thresholds,
instead of a single general threshold. Thus, the main focus of this work is on establishing an upper limit on the macroscopic,
or society level critical mass required to produce a social tipping event. Or the worst case scenario where social tipping can
still occur. Secondly, we would like to confirm any existence of a non-linear, or Pareto effect in social systems, where a
small minority can cause system wide social change. Although arguably only a general term for non-linear effects in social
systems, in the context of this research, it specifically denotes the potential of a small minority of people, roughly 20%, to
affect system wide change, to roughly 80% of the population. It is a well-known term in many spheres (Dunford et al.,
2014), and thus a good concept to communicate what is otherwise quite technical knowledge to a non-scientific audience.
Lastly we would like to more strongly bridge the conceptual and terminological gap between network science and social
tipping literature. Using an initially broad scope, we identify critical factors influencing tipping processes in social systems.
With these as a guide, we qualitatively review each factor and synthesise the existing information from relevant literature in
section 2. In the next sections, we then limit our analysis to literature which explicitly incorporates networks in a formal
sense, and include only those with empirical results. Finally, we relate our findings to social tipping in a more concrete and
applicable fashion in section 4.3. Our goal is to provide a first attempt at a broad-scope quantitative review of factors
influencing tipping processes, and ultimately to elucidate the realm of possibility in which tipping is most likely to occur.

## 2 Narrative literature review on tipping in social networks

### 2.1 What is social tipping?

Tipping refers to phenomena where a relatively small change or intervention in the system can lead to large system changes
in relevant system properties on the macroscopic level (Milkoreit, 2023). The term tipping point originates from social
science research on racial segregation patterns (Grodzins, 1957) and was used to refer to the thresholds in the racial
composition of neighbourhoods in the U.S. in the 1950s. Crossing those thresholds meant people with the skin colour that
was in the minority started to feel uncomfortable and tended to move out. More recently, the term was popularized by





Gladwell's (2000) book on trends in human behaviour and consumption, and technology change. The definition of tipping elements originates, however, in the work on the Earth's climate system (Lenton et al., 2008). The notion of tipping points and tipping elements since that time started to be broadly used in various scientific disciplines covering both the natural (e.g Holland et al., 2006; Scheffer et al., 2012; Dakos and Bascompte, 2014) and social sciences (Grodzins, 1957; Schelling, 100 1971; Milkoreit et al., 2018) (e.g. Grodzin 1957; Schelling 1971; Rogers et al. 2005; Doyle et al. 2016; Milkoreit et al., 2018). A formal definition of social tipping was proposed by Otto, Donges et al., 2020. The authors put forward that "tipping" involves a discontinuous state transition in the underlying system, i.e. it is more than a rapid continuous change. A key concept is the relative rate of change. The emergence of the new state, however, can be gradual.

105

The state of terms and definitions in this interdisciplinary field of social tipping research is quite inconsistent. There are mixed meanings, and terms are appropriated and in the process slightly change their meaning. Hence we spend some time defining, or re-defining concepts for the purposes of understanding in this article. A fundamental idea of social tipping (and tipping in general) is the idea of non-linearity, or a non-linear increase in a system state variable for a given increase in a 110 system control parameter. A useful and intuitive concept of "spillovers" to depict this comes from Efferson et al., 2020. A spillover is an indirect systemic effect produced by an endogenous response to an intervention on a single, or few individuals. This is larger than the effect of the intervention itself. In this sense, an increase in a system control parameter or the actions of a single individual would be an intervention, and the non-linear response in the system the spillover. The term spillover is intuitive when considering dynamical social systems. If we plot the steady-state value of the state variable $F(\lambda)$ as 115 $t \rightarrow \infty$ against a given value of the critical parameter $\lambda(t)$, and shade the region above $F(\lambda) = \lambda(t)$, this shaded area represents the domain where the system exhibits a nonlinear response, specifically where $F(\lambda) > \lambda$. We depict this concept in Fig. 1 below, where the blue shaded area is a "spill" over the reference line.



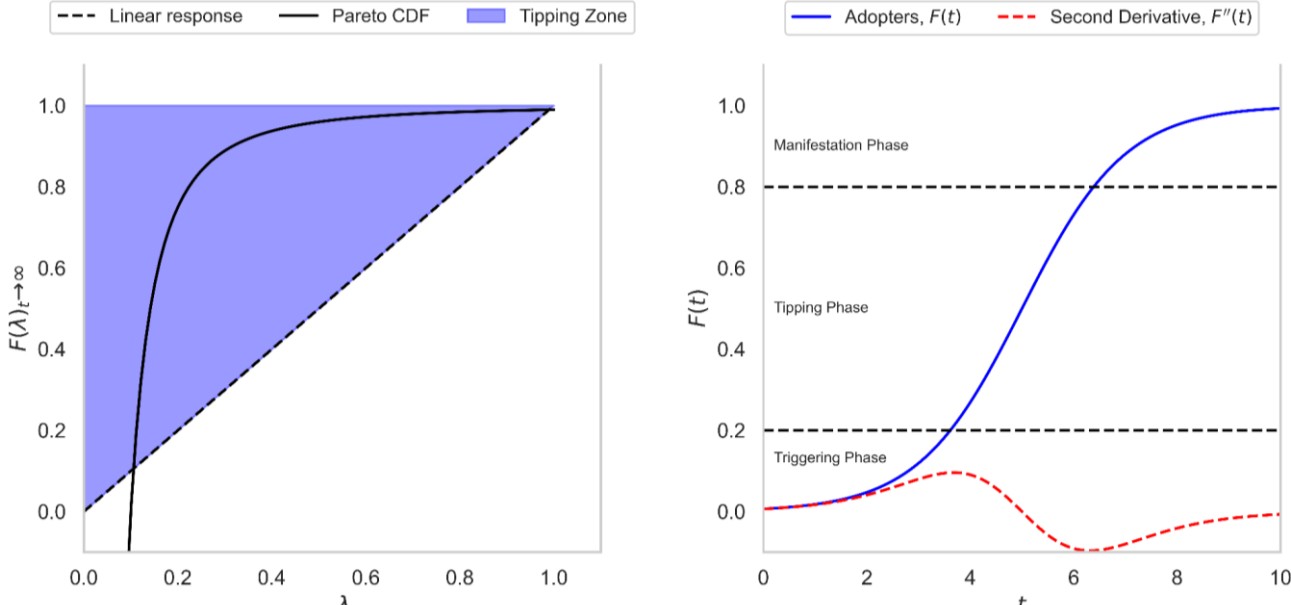

**Figure 1: (a) Depiction of the zone of non-linear outcomes based on the critical parameter value $\lambda$, denoted by "Tipping zone" and shown in blue. An exemplary cumulative distribution function depicts the Pareto effect where roughly a 20% change in a system control parameter results in an 80% change in the system state at equilibrium. (b) A possible time evolution of the fraction of adopters of a new norm in a social system undergoing a tipping event. We use these two concepts as a conceptual base for our review, and its methodology.**

Although this is a general example, if $\lambda$ represents the proportion of actors engaged in one social norm, and $F(\lambda)$ the final fraction as the time of the system approaches infinity, the shown Pareto CDF depicts a scenario in which a minority of actors have convinced a large majority to switch to another social norm. This is also what is referred to as a contagion event, or a cascade in network theoretic terms. Another way to view social tipping which will be employed in our analysis further on, is to look at the number of people as a fraction of the social system who have adopted a certain norm over time. A tipping point then can be identified as the point in time where this fraction $F(\lambda)$ demonstrates the most rapid potential for change. How do we define this? One option is to consider the concept of criticality. Here Smith et al., (n.d.) define this as the probability that the state of a complex adaptive system at a given point in time undergoes a tipping process, or a path bifurcation. We conceptualise this simply for our analysis with the second derivative of our state variable or the fraction of norm adopters in time. If we assume this rate is proportional to the probability of a marginal norm adoption per time, we can say that near a tipping point, it is also proportional to the criticality of the system. Jin and Yu, (2021) also adopt this measure to classify the tipping threshold of a networked social system under complex contagion. Which is classified as the chance of tipping based on a perturbation, or marginal (individual) norm change. In figure 1.b we plot this fraction $F(t)$. We apply the language from Otto and Donges here to conceptualise these stages (Otto, Donges et al., 2020).



---

**Box 1: Key Terms**

**Tipping event**

Assuming a social system where an agent can adopt norm states *a* or *b* at a given time, this pertains to the steady state fraction of individuals who have adopted norm *a*. The condition which is satisfied by a tipping event is defined as: *M(0) < 0.5* (indicating that norm 'a' starts in the minority) and *lim(t→∞) M(t) > 0.60* (signifying that norm 'a' becomes the majority over time). Where *M(t)* is the fraction of norm *a* adopters at a given time.

**Network cascade**

Analogous to a tipping event but on a network: A change in the behaviour of individuals (nodes) in a population (network) due to a herd-like behaviour through imitation of others. Subject to the *cascade condition*: An innovator or seed node has to be attached to a vulnerable cluster of nodes who become adopters, which after a percolation (spreading) process must occupy a fixed fraction of a finite network (Watts, 2002).

**Tipping threshold - Macroscopic**

Given a social system, refers to the point *c(t)* in the trajectory of *F(t)*, where *F(t)* represents the fraction of individuals in a social system who have adopted a certain norm at time t, whereafter a rapid and non-linear increase occurs in *F(t)*. Also referred to as a tipping point. See figure 1.b for a graphical example.

**Critical Mass**

The fraction of individuals F(*t*) in a social system who have adopted a certain norm at the time where the macroscopic tipping threshold is reached, represented as *F(c(t)) = λ*.

**Threshold fraction - Individual**

Given a node i in a social network: The fraction *ϕ* of network neighbours *k* of node *i* sharing a common state, after which exceeded, node *i* also changes their state.

---

**Box 1: Some key terms which are introduced by us and are helpful to understand the concepts in this article. Definitions may be similar to other works but are slightly changed to be applicable for our analysis.**

## 2.2 Networks and Tipping

Social processes are governed by the relationships between people. The spatial and temporal sum of these connections constitute social networks. In this sense network structure is fundamental to flows on a social network and critical for tipping





processes (Dodds and Watts, 2004). A formal description of networks is almost always the mathematical concept of a graph. In their most simple form, networks consist of nodes and links (Berner et al., 2023). Thus a network $N$ can be fully described by the tuple $N = (V, E)$, where $V$ is the set of all nodes, and $E$ is the set of all links. Here nodes can be people, animals, or molecules, and the links can be Facebook interactions, mating relationships, or bonds. Before giving concrete examples of

150 networks, it's important to distinguish between adaptive, temporal and static networks. Intuitively, the former two change their structure with time, the latter does not (Holme, 2015). Adaptive networks and temporal networks both shape, and are shaped by dynamic processes occurring on them, but the topology in the former takes precedence over the temporality, or timing of events (Holme, 2015; Berner et al., 2023). Considering that all social networks are predicated on social interaction, and thus are constantly changing, static networks are for all intents and purposes either representations of aggregated social

155 interactions, or network processes such as rewiring over a time period (time-aggregated networks, or a static slice of a network at some fixed time. A concrete example of a social network would be attendees of a conference. In this case each user is a node, and conversations between attendees represented as links (contacts) between them - a human proximity network (Holme, 2015; Donges et al., 2021). The sum of all conversations over the conference period, or a snapshot of those currently conversing at 15:00 on a Friday afternoon would then be a static representation. A temporal or adaptive

160 representation is more difficult and could be plotting the average degree of the graph against time (Holme, 2015) . In this work we consider all three types of networks, however the majority are either static or adaptive networks. Most literature, especially those involving modelling, use archetypal networks which are representative of commonly occurring real world networks and their properties. One example are small world networks, which possess the properties of high local clustering of nodes, and short path lengths which are often displayed by real world biological, ecological, and social systems (Watts

165 and Strogatz, 1998; Telesford et al., 2011) . A figure and reference of the most common networks appearing in our review is given below.

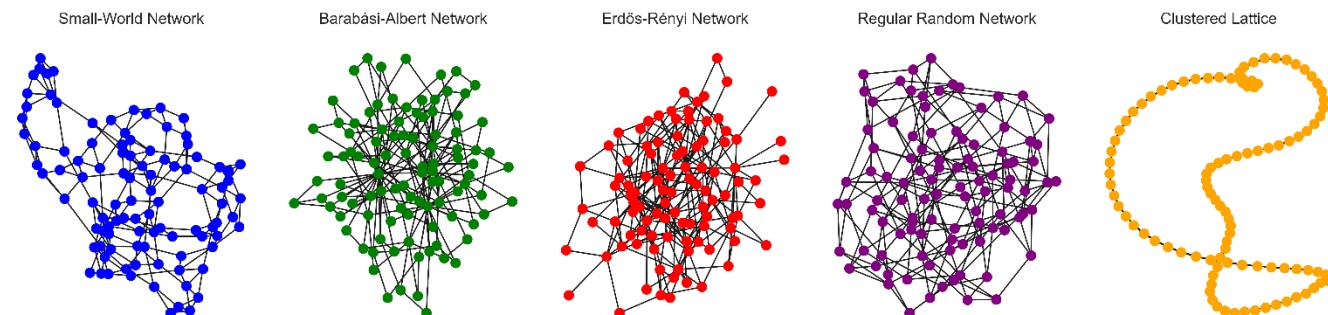

**Figure 2: A visual representation of the most common network types identified in our literature review.**

| Network Type | Clustering | Average Path Length | Degree Distribution |
| --- | --- | --- | --- |
| Small-World | High | Short | Varies |
| Barabási-Albert | Low | Short | Scale-free (Power law) |



| Erdős-Rényi | Low | Varies | Binomial/Poisson |
| Regular Random | Low | Long | Uniform |
| Clustered Lattice | High | Long | Uniform |

**Table 1:  A description of network characteristics for the networks shown in Fig 2.**

For the purposes of this work, which seeks to frame social tipping through the lens of network-theory, we can generalise social tipping as a contagious spreading process on a complex network, or a cascade (Guilbeault et al., 2018). A definition is given in Box 1. This spreading can involve behaviours, opinions, knowledge, or social norms (Christakis and Fowler, 2007; Nyborg et al., 2016; Schleussner et al., 2016). Using the language of systems-dynamics, the control parameters we are observing in this study relate to network properties or the properties of agents embedded in the network, for example thresholds, or even the adopting fraction itself at any given time, and the steady system state variable is the fraction of individuals who have adopted. In particular, where we are interested in finding the point of critical mass which facilities a tipping event or; the rate of change of cooperation with respect to the current level of cooperation or adoption, we can use Centola's 2013 definition of a tipping point:

"The gap between zero cooperation and the level of cooperation at which the growth of participation becomes self-sustaining."

In the rest of this article we will use the terms cascade and social tipping interchangeably, where we imply they are the same concept when discussing social opinion and norm dynamics on networks. We consider social behaviour to be an expression of the adherence or adoption of the norm, and use the terms interchangeably. An important distinction regarding thresholds is to be made between the system level macroscopic tipping threshold and individual agent thresholds, or threshold fractions. Where the former is defined as in fig 1.b, also in box 1 as the tipping point in a time series, the latter refers to the conditions in an agent's immediate social network required for one agent to change their opinion (Watts, 2002). The difference is important as under most realistic cases, the mean individual threshold will not be equal to, nor reliably predict a given macroscopic threshold (Wiedermann et al., 2020) .

**2.3 The role of network structure and attributes**

In this section we examine the effects of some network traits or properties on tipping processes on some well-known network topologies. However not all networks are the same, topology and thus structure can vary based on which social domain (Efferson et al., 2020), social group (Christakis and Fowler, 2008) or social process (Bellotti et al., 2023) the network represents. Whether the network varies with time, or is shaped by that very social process, as occurs in temporal and adaptive dynamical networks (Berner et al., 2023). These factors have effects on outcomes of processes on networks, and are often confounded with one another (Shalizi and Thomas, 2011). For example, financial networks display more inequality in





degree distribution than a reference small world network (Leo et al., 2016), homophilous networks spread health innovation behaviour more effectively than unstructured networks (Centola, 2011) and bursty network interactions can allow contagion events in networks which are otherwise difficult to tip (Karimi and Holme, 2013).It can be difficult to address the role of

network structure when most of the networks we are discussing in this work are essentially adaptive dynamical networks, i.e. have constantly evolving structure. With this in mind, this section will address how this static structure affects cascade dynamics around a certain time point.

Focusing on well-known network topologies has the advantage of avoiding confusion of terms between fields, where certain

network types are ubiquitous, e.g Erdős-Rényi, Barabasi Alber (Albert and Barabási, 2002), or Watts-Strogatz (Watts and Strogatz, 1998) networks (Telesford et al., 2011). There is broad evidence of common relationships between network topology, cascade size and frequency. For example, evidence from game theoretical (Ohtsuki et al., 2006), ecological (Martin et al., 2020), as well as social contagion based models (Centola, 2011, 2013) all show that a structured network is beneficial to the magnitude and rate of contagion spread in comparison to unstructured networks. Although in the former it

is the spread of cooperation. This is in contrast to the "strength of weak ties" concept demonstrated by Granovetter (1973) and others (Watts and Strogatz, 1998). The answer to these contradicting results is that it depends on network size. Centola, 2013 demonstrates how for small systems weak ties are mildly helpful in contagion spread, but for larger systems, strong ties and clustered networks are required for successful critical mass phenomena. Where social tipping for sustainability plays out on a global scale, a prerequisite for any mobilisation is therefore the existence of homophilic, interconnected and trusting

networks. Although this is generally the case, Efferson et al., 2020 show how homophilly can be detrimental to spillovers in the context of policy interventions, when they are too large. This implies that attempts to facilitate norm change exogenously may interact with homophily in detrimental ways once the intervention becomes too strong. Clustering, more specifically, increases the likelihood of repeated exposures to a contagion source, and acts to lock the information in a community (Fink et al., 2016). This second part is fundamental for the formation of a critical mass (Centola, 2010) halting the dispersion of a

social contagion for long enough for a percolating cluster to form. Overall complex contagion requires that the network has communities which are sufficiently built up to allow ideas to reinforce themselves within, but enough connectivity or long ties that these similar clusters can connect at some point (CHIANG, 2007). This idea is fundamental in the ever more highly connected global networks we find ourselves living in, where information supply is higher than ever, as well as noise (Bak-Coleman et al., 2021). Contagion, or the information about it tends to die out after more than 3 network steps (Christakis and

Fowler, 2007, 2008; Fowler and Christakis, 2008), hinting that there is some fundamental laws that govern network structures which are compatible with these stickier cascades we are dealing with.

## 2.4 The role of actor's preference and heterogeneity

A fundamental requirement of successful emergent social tipping processes is that consecutive individuals or agents are susceptible to change. Many terms are used to describe this concept of preference for change, however in models of norm




change or opinion spread across disciplines, it is often operationalised implicitly or explicitly as a threshold (Efferson et al., 2020; Centola, 2013; Guilbeault et al., 2018; Granovetter, 1978; Watts, 2002). In these models, a threshold quantifies the point at which an agent will change their behaviour and thus governs the magnitude and rate of a cascade, or tipping, in a population. In the real world, this preference varies individually (Efferson et al., 2020), and is also highly dependent on the type of normative change (Guilbeault et al., 2018; Berger et al., 2021). In other words, not only are individual thresholds

heterogeneous, but so too are their governing distributions. Macroscopic or social-group-level threshold distributions are also emergent, meaning that their shape is not visible or predetermined, but arises out of a unique set of interactions between microscopic actors (Wiedermann et al., 2020). This property makes prediction exceedingly difficult, especially in highly polarised or controversial issues. Wiedermann et al., 2020) successfully demonstrate how agents seeded with very narrowly distributed individual (fractional) thresholds can produce a different looking system level distribution. A number of models

and experiments show the significant effect different threshold distributions have on both cascade speed and magnitude (Andreoni et al., 2021; Berger et al., 2021; Karsai et al., 2016; Dodds and Watts, 2004). Efferson et al., (2020) demonstrate how this effect is also robust to changes in network topology, intervention types, and several other factors. Individuals with high thresholds or even untippable or "immune nodes" for a given spreading event can severely hinder or prevent a cascade process (Karsai et al., 2016; Wiedermann et al., 2020). This potential effect is magnified when these nodes occupy key

positions in a network, for example as first contacts of an innovator, or a seeder for a potential network contagion. Where optimally, this first contact network should consist of individuals with typically lower thresholds than normal to enable cascading (Nishioka and Hasegawa, 2022). Efferson et al., (2020) also specifically show that under some conditions (where a positive response to an intervention is guaranteed), targeting resilient nodes with policy interventions is more effective than leaving norm change up to endogenous processes such as tipping or spillovers.


Thresholds are influenced by a number of often co-dependent factors, examples are; payoffs or switching incentives (Centola et al., 2018) tension (Berger et al., 2021) and jointness of supply (Centola, 2013) to name a few. These terms all refer to a switching-payoff, or the cost of norm adoption (abandonment) but are expressed differently. They in turn are all dependent on network density, social context, and type of norm change (Berger et al., 2021; Efferson et al., 2020). Perhaps confusingly,

these terms are also used in some models as implicit thresholds, for example in (Andreoni et al., 2021) where tipping thresholds are set through changing miscoordination penalties, or increasing the personal benefit of change. Conversely, explicit thresholds are used to operationalise these same concepts. Examples are seen in Berger et al., (2021), and Efferson et al., (2020), where different threshold distributions are used to represent different social preferences and tension related to a specific dilemma. Following this example, Fig. 3 displays several general distributions which may represent preferences via

tipping thresholds for certain socio-ecological dilemmas. An example is reducing





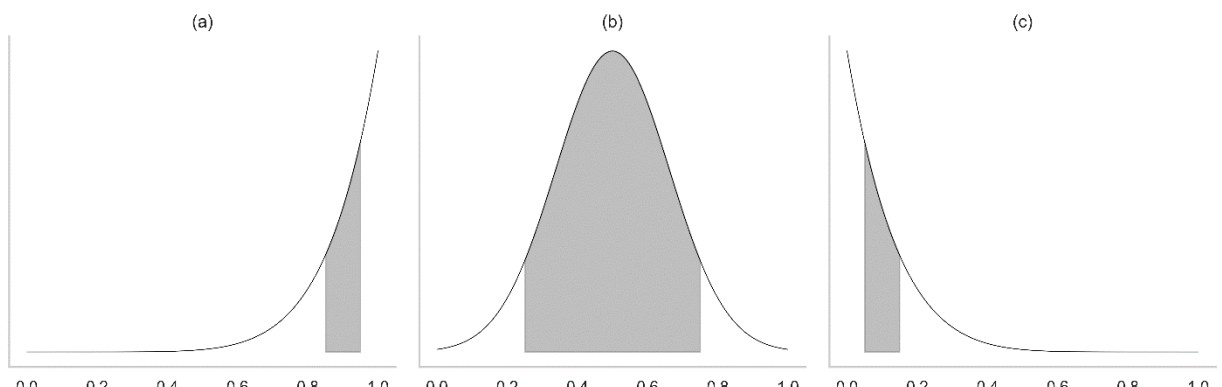

**Figure 3: Illustrative threshold fraction ($\phi$) distributions for a population. These indicate preferences toward changing a specific behaviour or norm in reference to some current social dilemmas surround pro-environmental behaviour. Here, (a) could represent a decision to go vegan, (b) to ride the bike to work, and (c) to recycle rubbish and waste. The shaded area represents different strategies for choosing members of a network, the seeds, to try and facilitate endogenous norm change on a network, or social tipping.**


All meat and animal product consumption, which would correlate with a left skewed distribution (a), meaning that the mean threshold is high, and tipping is difficult (Peattie, 2010a). In this situation, most people would only change dietary habits

when a vast majority, around 70%, are consuming food differently. Intuitively such a dynamic precludes the existence of any tipping dynamic, as the relationship of tipped elements to chance of tipping is 1 to 1 or linear, for the majority of the tipping window. Another is car usage reduction, whose controversial and polarised nature may be represented by a bi-modal distribution. In this case, willingness to drive less is strongly very low in one half of the population, and very prevalent in the other. Regardless of nomenclature, a number of sources show the successful adoption of a cascading norm or behaviour is

highly contingent on the perceived individual benefit, regardless of the magnitude of the cascade (Centola, 2013; Centola et al., 2018; Berger et al., 2021).

## 2.5 The role of Agency and inequalities

In this section we ask how individuals and groups can intentionally influence the adoption of new patterns of behaviour (Kaaronen and Strelkovskii, 2020) and induce abrupt changes in social conventions and public opinion (Centola et al., 2018;

Galam and Cheon, 2020) Specifically, what is the agency of individuals and groups in transforming the social structure understood as the collective prescriptions and constraints on human behaviour (Granovetter, 1985; Robb, 2014) The social structure is composed of the rule system that constitutes the "grammar" for social action that is used by the actors to structure and regulate their transactions with one another in defined situations or spheres of activity. Burns and Flam, 1987, p.26 point out that the complex and multidimensional normative network is not given, but is a product of human action;

"human agents continually form and reform social rule systems". However, what is the agency of individuals to change the social structure? Human agency is understood as the ability to shape one's life and a few dimensions can be distinguished. Individual agency is reflected in individual choices and the ability to influence one's own life conditions and chances. This



individual agency varies strongly within the society across age, gender, income, education, personal health status, position in social networks, and other dimensions. Collective agency refers to situations in which individuals pool their knowledge,
skills, and resources, and act in concert to shape their future. Everyday agency refers to consumer and daily choices, and strategic agency refers to the capacity to affect the wider system change (Otto et al., 2020a)

In a network-theoretic sense, agency can be seen as the ability for a node to control or initiate processes on a network. Where structural properties of a network, or a node, such as centrality, or degree strongly influence this ability (Korkmaz et al.,
2018), we can use these structural measures as a proxy for a node's agency. Structural properties, while generally a good indication of a node's influence, do not tell the whole story. The agency of a particular node is also dependent on the specific dynamics occurring on a given network, and their context. This is clearly demonstrated in Guilbeault and Centola, 2021, who show that standard centrality measures, while suitable to predict the social influence of seed nodes under simple contagion dynamics, fail under complex contagion. Social influencers, who in colloquial terms have high agency as per our definition
above, have been the subject of much contention in several areas dealing with research on social change (Paluck and Shepherd, 2012; Paluck et al., 2016; Bellotti et al., 2023; Watts and Dodds, 2007; Nishioka and Hasegawa, 2022; Nyborg et al., 2016; Hodas and Lerman, 2014; Han et al., 2020; Nielsen et al., 2021). An intuitive view of social influencers, and their presence in the era of social media platforms such as TikTok and Instagram could lead to the belief that they may dramatically shape social opinion and information. However, in the world of complex contagion they may be surprisingly
ineffective at affecting social contagion processes (Watts and Dodds, 2007). In fact "normal" people may be the most cost-effective instigators of change, especially as the volume of information reaching us increases more and more (Bakshy et al., 2011; Fink et al., 2016; Hodas and Lerman, 2014).

The question is how in situations in which individuals and groups with different and conflicting interests the change takes
place. Centola (2021) points out to the role of so-called change agents, who bring innovative solutions in their communities, advocate changes, build networks of early adopters, and play a pivotal role in coordinating on the new equilibrium and restructuring institutions.

## 2.6 The role of processes, time, theme, and scale

Processes which imply a temporal element have a large effect on social cascade dynamics and due to the interdependence
between processes, network structure, and agent state variables, they can be difficult to disentangle from one another as mentioned in 2.3. Some sources claim that temporal considerations can be more important than network structure, or can simplify some aspects of complex spreading (Karimi and Holme, 2013; Hodas and Lerman, 2014). In the former study, the duration over which interactions occur strongly affects cascade magnitude and success. Perhaps to highlight the difficulty of making general statements in these system, the duration length shows the opposite effect depending on whether a fractional
or absolute threshold is used in the cascade model. Information transmission rate or bustiness can be conducive to complex



contagion (Karimi and Holme, 2013), where in contrast it's shown to slow down simple contagion (Karsai et al., 2011). Information about the social norm landscape, both globally (norm average), and locally (close contacts) strongly influences the decision to abandon an old norm or adopt a new norm (Bergquist and Dinerstein, 2020; Pieters et al., 1998; Leviston and Uren, 2020) This information may pertain to the prevalence of a social norm in society, and is very important when the perceived risk or change is high (low payoff). For example when a person is deciding to abandon a behavioural norm but faces the penalty of alienation from their close social group. When this agent knows that despite the group norm, there is global support for an alternative norm, they may be more encouraged to switch regardless. Andreoni et al., 2021) confirms a general effect to this extent in a behavioural experiment, where providing participants with information about other players preferences they were not directly linked to increased contagion size. (Jin and Yu, 2021) also show a similar effect in a modelling approach. This is a key factor when considering something like pro-environmental behavioural norm changes involving for example eating less meat (Leviston and Uren, 2020), where the risk of alienation is high. Information frequency, or regularity, and clarity, is then an essential part in ensuring cascade events are noticed by people in a social network, essentially increasing the fraction of people available to engage with a cascade. Irregular or delayed belief update times, as well as unclear information work to dampen cascades, and prevent the formation of a critical mass, as people become risk-averse under poor information (Berger et al., 2021; Peattie, 2010b). As a caveat, when the information density, which may be considered as information frequency per time, becomes too low, social contagions may fail to infect a person, as they do not attach enough importance to the information, or do not notice the signal (Hodas and Lerman, 2014). This can also be thought of as a poor signal to noise ratio. Ref identifies this as one factor which makes nodes with a high in-degree, common with social influencers, more difficult social contagion targets than others. They are overwhelmed with noise. To a lesser extent, the general noise created by our highly-interconnected digital global network may make complex contagion difficult through these mediums (Hodas and Lerman, 2014; Bak-Coleman et al., 2021).

We established earlier that norms and opinion spread differently to viruses and memes for example, and these can be roughly separated into complex and simple contagions respectively. This simple dichotomy hints at a fundamental principle that every type of contagion may spread differently. Indeed as an example, Christakis & Fowler (Fowler and Christakis, 2008; Christakis and Fowler, 2008, 2007) in their long term study of a network of 12067 people over 32 years show that the spread of happiness is more dependent on a person's geographical proximity to a potential contagion source than the spread of healthy eating behaviour. Smoking behaviour transfers very easily with one's spouse, but not obesity, or happiness. Finally, educated people in the USA will have more influence on others' smoking behaviour, but in another study amongst rural communities in India, local wise-people and knowledge holders only have a marginal effect on the spread of malaria-prevention behaviour (Bellotti et al., 2023). Norms related to controversial topics such as politics, or social movements in response to socio-political issues show large marginal effects after continued exposure to a norm holder, showing that repeated exposure is critical for opinion change (Romero et al., 2011; Fink et al., 2016). This unique variation in spreading behaviour based on content can make prediction even more difficult. It is noteworthy that all the studies mentioned above



still report repeated exposure and social proximity as a leading predictor of norm spread between people, which lends support to the use of complex social contagion models even in unfamiliar contexts or under uncertainty.

## 3. Data and Methods

### 3.1 Data collection

To identify literature regarding social tipping in networks from across various disciplines, several broad search terms and
strings were initially used, as many disciplines employ different nomenclature for non-linear changes in their respective approaches. Where we are explicitly focusing on networks, we include this in every search string. A literature search was conducted on Web of Science, as well as Google Scholar for the period 01/01/2001 - 20/09/2023. Search terms used were "complex contagion" AND "social networks", "network"*, "networks"* AND ("complex contagion" OR "norm diffusion"), ("norm diffusion" OR "complex contagion") AND "social networks". The search results were used to identify 33 studies
using modelling, observational or experimental methods and mentioned or referred to empirical results in their abstract. A further 27 were discovered by going through reference lists of the initially identified literature, and using comprehensive review articles recently conducted on complex contagions (Guilbeault et al., 2018; Holme and Rocha, 2023). Of the 60 studies identified, 21 had to be discarded as on further inspection they did not include complex contagion models in their methodology. We proceeded to analyse the final list of literature based on stages: In stage 1, key empirical results are
elucidated and coded into a database, stage 2 includes the evaluation of these key results and relevant theory which is synthesised in section 2. In this section we also draw on literature in the reference material of the primary literature to bridge knowledge gaps and supplement our synthesis. This material is not included in the database, but can be found in the references as usual.

  The number of literature considered in these stages was N = 41. In stage 3 we filter the literature so that only those with
quantitative results which are of a similar dimension, or can be rescaled to be so are kept. At the end of stage 3, we were left with N = 12 articles. Stage 1 and stage 3 results are displayed in sections 4.1, and 4.2 respectively.

Stage 1 involves the classification of key results in terms of how they influence cascades on networks. Concretely, the two criteria we judge upon are the effect on the rate and magnitude of the cascade. We use a simple grading system to ease data
collection shown below in Table 2. Where many of these effects display non-monotonic behaviour, we code them accordingly and they are represented on the x axis in Fig. 5 as "+/-". For results which cannot be quantitatively graded, we simply mark them as having a positive or negative impact on the cascade likelihood. This can be interpreted as an increase to the probability, speed, or magnitude of a cascade event. Stage 2 involves a more involved evaluation of key results to find overlaps and agreements between fields. Where terms have conceptual or mechanistic agreement, and are used in the same
context, i.e. the study was evaluating a particular aspect of their effect, we group them under an overarching, higher level term. An example are the concepts of rewiring (process), awareness of others preferences (process), and weak network ties



(structure). All seek to increase the distribution of information around the network to agents and are classified under the term "global information". A glossary of terms and their meaning can be found in Appendix A, table A1.

| Percent change | Positive Impact (+) | Negative Impact (-) |
|---|---|---|
| 0-30% | 1 | -1 |
| 30-60 | 2 | -2 |
| >60% | 3 | -3 |

**Table 2: Categories of grouping terms based on a percent change to the magnitude of a tipping event compared to a baseline scenario.**

### 3.2 Intercomparison of tipping data from Models and Experiments

In order to quantitatively compare tipping data across compatible literature sources, we recovered 8 modelling datasets and 4 experimental datasets either from contacting the respective authors, retrieving published data, or re-running simulations

based on software published with the articles. For literature where neither of these things were possible, trajectories or data were extracted directly from articles using optical character recognition (OCR) or other graphical techniques. All the models evaluated included complex-like contagion dynamics regardless of the technical implementation, i.e. even if models didn't explicitly use a contagion model, the social spreading dynamics included a threshold-like mechanism of contagion, where agents needed multiple exposures to be infected. As mentioned in section 2.1, the focus of this review is primarily the

macroscopic tipping threshold, as it allows us to bound our analysis and compare units more easily across studies, as most literature reporting qualitative results includes a time series. This is helpful where parameter dimensionality can be very high and its overlap low. Assuming a time series of the fraction of adopters in a given dataset is present, we calculate the tipping threshold $c(t)$ from each. This is considered to be the point where a sufficient fraction of agents in a social network sharing the same state value has been achieved in order to facilitate a rapid cascade (defined as over 50%) of state value changes

over the rest of the susceptible network population. We refer to the critical mass at the tipping threshold $c(t)$ as $\lambda(t)$. We drop the time dependency in notation going forward for brevity. We find the critical threshold $c(t)$ as defined in section 2.1: i.e the point in the time series of F(t), the fraction of adopters of an alternative norm where the criticality (see section 2.1) in the system is the highest. More concretely we assume this to be the point in $F(t)$ where the second derivative reaches its maximum. This can be expressed as:

$$c(t) = max_t(F''(t)). \tag{1}$$

Where trajectories are non-continuous, as in experimental results, finite difference methods are employed to estimate $c$. To simplify our results, we limit our quantitative analysis to systems which demonstrate a successful cascade. Assuming two possible norm states a or b, this pertains to the steady state fraction of individuals who have adopted norm a at time t, represented as $M(t)$. The condition is defined as: $M(0) < 0.5$ (indicating that norm 'a' starts in the minority) and $lim(t→∞)$

$M(t) > 0.60$ (signifying that norm 'a' becomes the majority over time). Where we are also interested in microscopic or



individual level thresholds (threshold fractions), we have collecting ranges of mean individual thresholds where a cascade event is possible. This measure allows us to compare the effect of thresholds on cascade success across studies using different threshold distributions, and shapes. Individual thresholds are compared as a threshold fraction in this work and defined to be the fraction $\phi$ of network neighbours $k$ of node $i$ sharing a common state, after which is exceeded, node $i$ also changes their state.


## 4 Results

Below we summarise the main mechanisms which affect social tipping success as identified by parsing qualitative results from the literature. A table of terms is provide in order to provide understand for the network abbreviations.

| Term | Abbreviation |
|---|---|
| Clustered lattice | CL |
| Erdős–Rényi network | ER |
| Regular random network | RRN |
| Small world network | SW |
| Holme-Kim network | HK |
| Scale-free | SF |
| Watts-Strogatz | WS |
| Power-law | PL |
| Barabási–Albert model | BA |
| Scale free | SF |
| Small world network | SW |
| Erdős–Rényi | ER |


**Table 3: A summary of network topology abbreviations for Table 4, below.**

| Citation | Network_Topology | Key_Mechanisms | Supplementary_Mechanisms | Publication_Year |
|---|---|---|---|---|
| Andreoni et al. (2021) | Empirical network | Switching payoffs; switching threshold; | Personal preferences; public awareness of preferences; Time-scale | 2021 |
| Amato et | Empirical(conversati | Policy (institutional | Informal institutions | 2018 |



| al. (2018) | on network | intervention); committed activists | | |
|---|---|---|---|---|
| Centola et al. (2018) | Complete network | Coordination payoffs; committed minority size | Individual memory length; population size | 2018 |
| Centola (2013) | CL, RRN | Jointness of supply (coordination payoff); homophily | network structure | 2013 |
| Baronchelli et al. (2006) | Complete network | System size | Scaling relations | 2006 |
| Xie et al. (2011) | ER, BA, complete network | Network topology | Immune nodes; critical minority size | 2011 |
| Castilla-Rho et al. (2017) | Grid | "zealots" - rule followers; group norm enforcement (pressure to conform) | Network connectedness - - average degree; group size | 2017 |
| Paluck et al. (2015) | Empirical (school) | Characteristics of seeds; out-degree of seeds | Zealots | 2015 |
| Wiedermann et al. (2020) | ER | Switching threshold distribution; fraction of acting individuals | Average degree | 2020 |
| Karsai et al. (2016) | Empirical (skype) | Immune nodes; switching thresholds | Constant flow of innovators | 2016 |
| Watts, Duncan J. (2002) | SF | Influence of seed nodes | Degree/threshold heterogeneity | 2002 |
| Faribi & Holme (2013) | Empirical (internet community), ER | Network temporality | Switching threshold | 2013 |
| Nishioka & Hasegawa (2022) | ER, empirical (facebook) | Switching thresholds; influence of seed nodes | Clustering; network typology | 2022 |
| Lacopini et al. (2022) | Empirical (various) | Social influence of seed nodes; stubbornness | Higher order network structures | 2022 |
| Krönke et al (2020) | ER, BA, WS, empirical (various) | Clustering; reciprocity | Network topology | 2020 |



| Karsai et al. (2014) | Empirical (skype) SF | GDP; press liberty | Network topology | 2014 |
|---|---|---|---|---|
| Barash et al. (2012) | Lattice, PL, SW | Long-range-ties; influence of seed nodes | Network topology | 2012 |
| Bakshy et al. (2011) | Empirical (twitter) | Social influence (spreader); url type | Content categories; interest; feeling | 2011 |
| Han et al. (2020) | PL, Empirical | Preferential contact of nodes (small vs large degree); information transmission | Population size; mean degree | 2020 |
| Jin & Yu (2021) | ER, BA, HK, lattice, SW, RRN | Global information; information sources | Network topology | 2021 |
| Zhu et al. (2019) | ER, ER-SF, SF-SF | Network heterogeneity | Threshold distribution | 2019 |
| Efferson et al. (2020) | Homophilus, complete, RRN | Switching threshold heterogeneity; coordination/switching payoff | Cultural identity; group norm | 2020 |
| Hisashi et al. (2006) | Lattice, SF, RRN, C | Ratio of payoff to degree; network topology | Population size | 2006 |
| Min & San Miguel (2023) | ER | Rewiring probability; network "plasticity" | Average degree | 2023 |
| Watts and Dods (2007) | RRN | Social influentials | Network density; network degree distribution | 2007 |
| Damon Centola (2010) | CL, SW | Homophily; network topology; exposure count | Clustering | 2010 |
| Damon Centola (2011) | ER, SF, SW, empirical | Homophily; network topology | Node centrality | 2011 |
| Gizem et al. (2018) | Lattice, ,SW, ER | Network structure; social influence (key nodes) | Clustering; degree distribution | 2018 |
| Okada et al. | Lattice, SW, RRN | Network structure; trust; density | Polarization | 2022 |





| (2022) | | | | |
|---|---|---|---|---|
| Ehret et al. (2022) | Complete graph | Group identity | Preference distribution; population heterogeneity | 2022 |
| Hodas et al. (2014) | Empirical(Twitter) | Social influentials; information density | Clustering; intensity of exposure | 2014 |
| Belloti et al. (2023) | Empirical(Twitter, Dig) | Frequency of exposure to contagion; household exposure; trust | Weak ties; social influentials | 2023 |
| Christakis & Fowler (2008) | Empirical(friendship, smokers network) | Trust; social proximity; social tie strength | Social influentials (education); clustering | 2008 |
| Fowler & Christakis (2008) | Empirical(friendship) | Trust; social proximity; social tie strength | Physical distance | 2008 |
| Christakis & Fowler (2007) | Empirical(friendship) | Trust; social proximity; social tie strength | Household contacts | 2007 |
| Centola & Baronchelli (2015) | Empirical | Network topology | Network size | 2015 |
| Bond et al (2012) | Empirical(Facebook) | Social tie strength; geographic proximity | Weak ties | 2012 |
| Fink et al (2015) | Empirical(Twitter) | Thresholds; clustering | 2015 | |

**Table 4: A summary of network topology, and the key and supplementary mechanisms which were identified to have an impact on social tipping events based on the results in each paper.**


Contradiction was common across the literature for a number of factors, which is expected given the nature of complex contagion on complex adaptive systems. To give an idea of the degree of heterogeneity, we counted $N = 36$ different network topologies, and $N = 22$ different population sizes across the scope of the reviewed articles. Several variables showed non-

monotonicity within models and experiments, which are designated by the "+/-" symbol in Fig 5. Some of the most divergent findings occur around topics such as homophily temporal dynamics of network processes, and network size are reflected in Fig 4, where several studies argue for both sides of its effect. Homophily, despite the contention between sources



is shown to have overall slightly more positive support in the literature and a strong positive effect on tipping cascade size in certain circumstances. Social influence, which was mentioned in concert with social influencers quite frequently in the articles is shown to have a positive, and substantial effect on contagion as is shown in Fig 4, and Fig 5. Important to note however that social influence is not the same as social influencer. Factors pertaining to social influencers are multiple and include a high in-degree, which is associated with a reduction of infection probability from a cascade for reasons mentioned in section 2.6. There was broad agreement across the literature that trust and clustering has a strong positive effect on cascade magnitude, as well as overall success. Together, clustering, social proximity and trust were the factors with consensus across the literature as to the sign of their effect. These factors all serve to increase the frequency or amount of exposures to close contagion contacts and thus help satisfy the fundamental requirements of complex contagion spread.

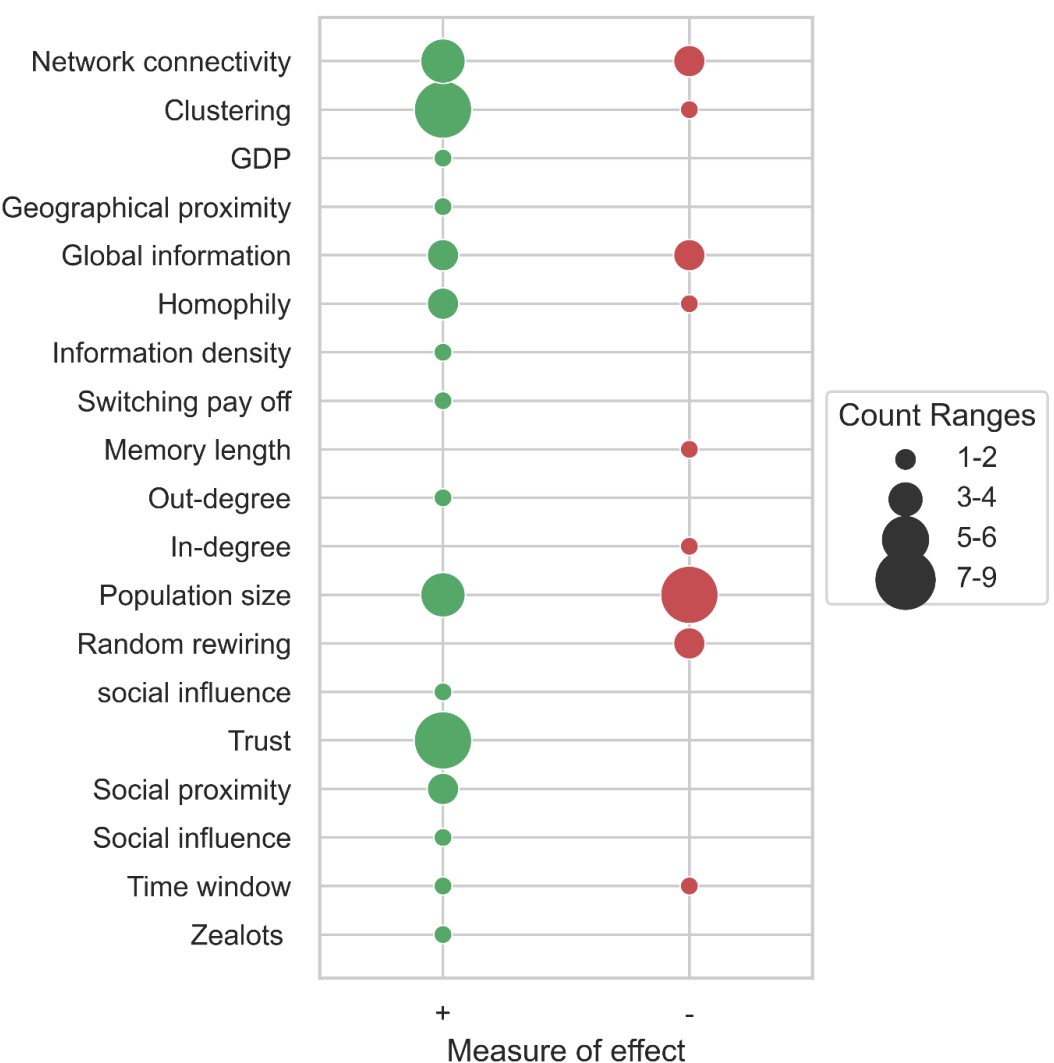





**Figure. 4: Frequently cited factors influencing complex contagion events in social networks. Summary of empirical results based on *n* = 91 observations over *N* = 39 studies. Some concepts have been harmonised over disciplines where compatible. Factors with**
**a sample size of 1 are not shown here to aid visibility but can be found in the SI. Population size, global information and temporal structure show high disagreement across the literature and are dependent on the context of spreading processes. Trust is a key factor in social complex spreading.**

Conflicting results should not be seen as arguments or weights for the absolute effect of a factor, but rather as a tendency, or

the probability of an effect to affect contagion. This pluralistic approach is necessary as most of the differences seen in

Figures 4 and 5 are due to strong contextual factors influencing the dynamics of the system in question.

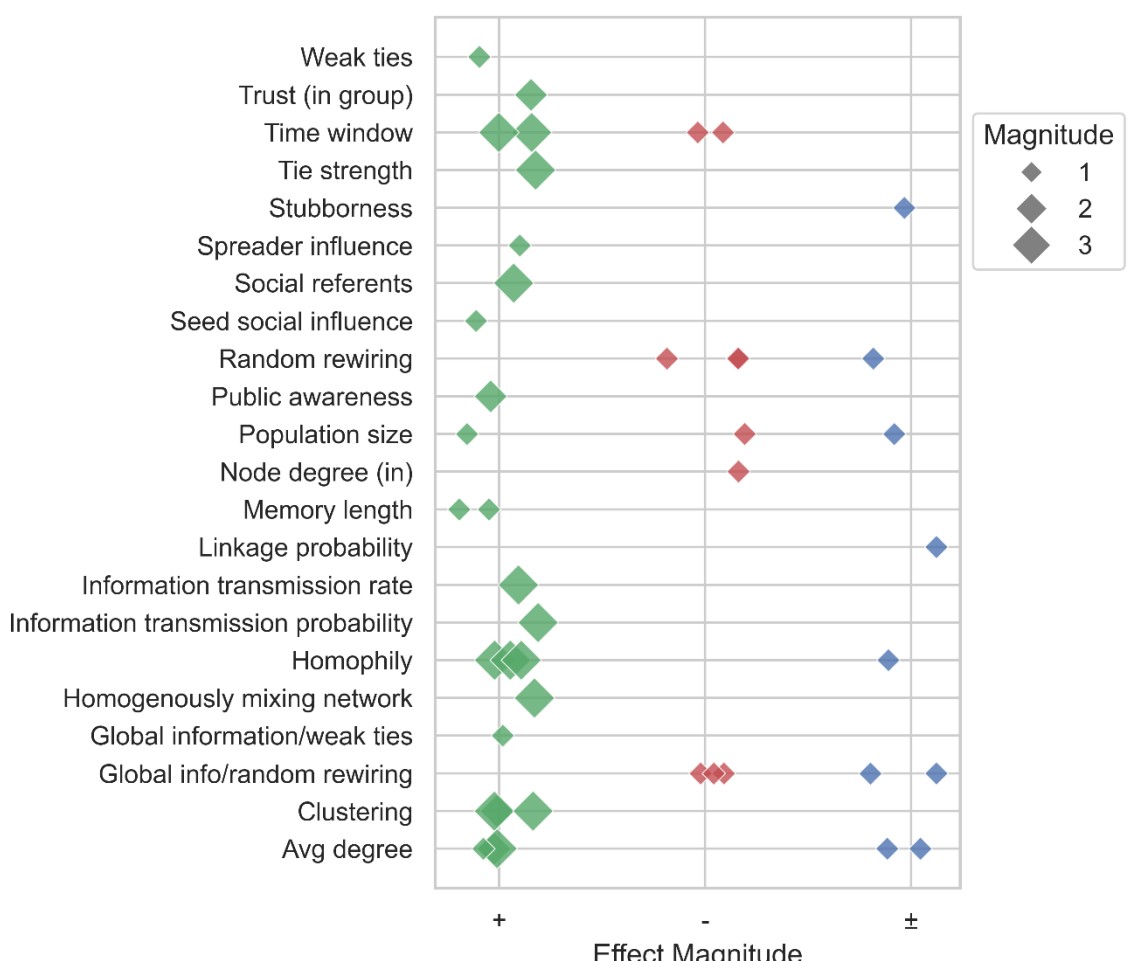

**Figure 5: Factors influencing the magnitude of contagion events in social networks. Values for literature with more discernible**
**data on effects, *n* = 50. Magnitudes are defined as per Table 2 and range from 0-100% impact on cascade magnitude. The relationship is displayed for an increasing value of the listed factor against a baseline scenario.**





Our analysis of critical mass sizes and the steady state adopter fraction as per Fig. 6 shows that for most social systems, there exists a critical mass of individuals who have adopted a norm after which, the system quickly shifts into a regime where all

other individuals adopt this norm. This occurs at approximately 25% of the total population size. This conforms to theoretical predictions about non-linear social processes, and may seem unsurprising that modelling results also replicate this. However what is interesting is that empirical results which categorise observational and experimental results are in general agreement with the modelling results, as well as each other. Empirical results tend to demonstrate sharper thresholds and non-linearity, verging on discontinuity. We see this effect also continuing across timescales, where for

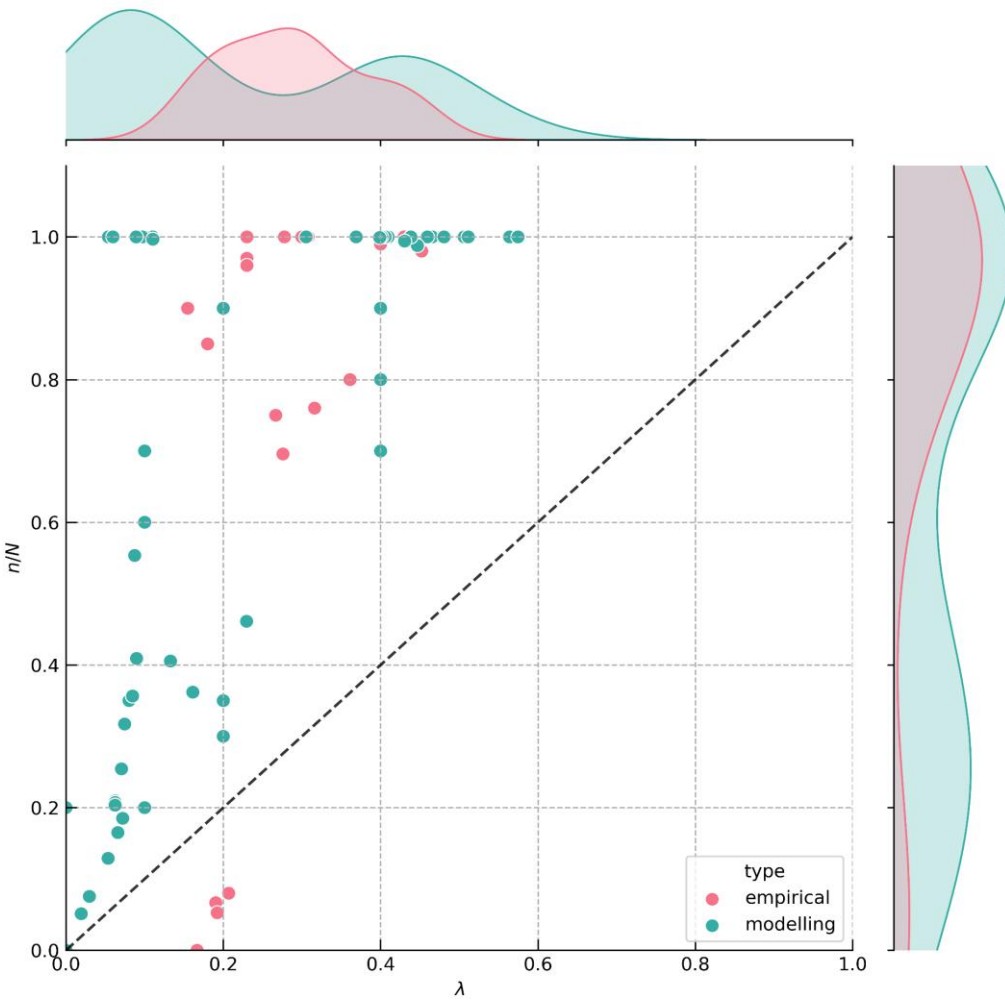


**Figure 6: The critical mass size λ and the steady state adopter fraction *n/N*. Here we show 87 modelling and empirical results of complex contagion in social networks. The roughly bimodal distribution of steady states as shown by the y axis marginal distribution, shown on the right, supports theoretical predictions about the non-linearity of social tipping. After a critical mass of ∼25% in a population, the steady state distribution of norm adopters converges quickly to a fully tipped state (*n/N* ≈ 1).**





example the results in Fig. 6 from Amato et al. (2018) have a time scale of centuries, to the days and weeks of behavioural experiments from Centola (2018), and Andreoni (2021). This implies a scale invariance in the tipping dynamics with respect to time. The bimodal distribution in the critical mass size seen on the top margin of Fig. 6 is likely a result of different modelling approaches to complex contagion. Some models inherently feature non-linear but continuous transitions to the tipped state, such as analytical approximation methods of Granovetter's tipping threshold model (Xie et al), whereas

numerical methods tend to show discontinuities. Certain functional forms representing tipping are also responsible, for example, system dynamics models using normal forms to model social tipping (Kroenke et al. 2022). These normal forms may inherently feature certain dynamics such as discontinuous bifurcations. Leading to the behaviour observed in Fig 1.

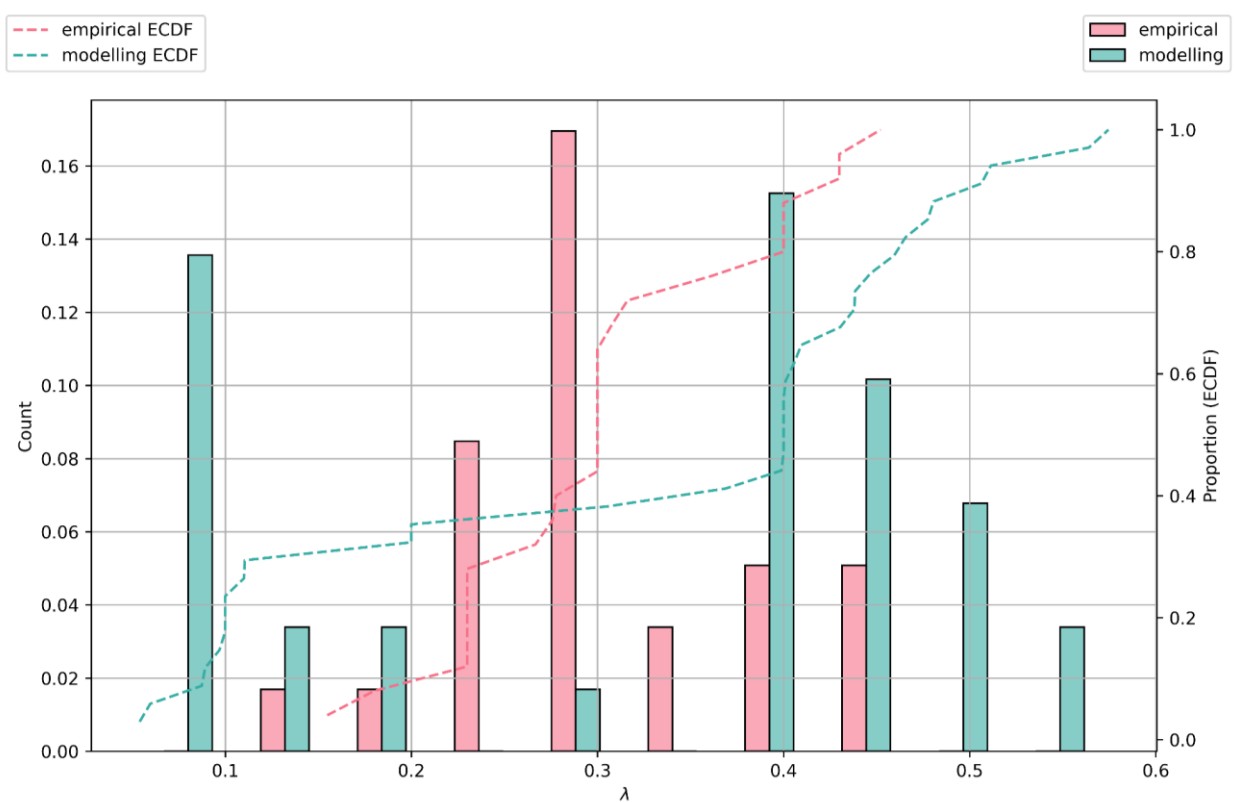

**Figure 7: The distribution of the critical mass λ. Here we classify only successful tipping events, i.e *n/N > 50%* of the population, numbering 59. The empirical cumulative distribution function (ECDF) demonstrates that the majority of critical masses conducive**
**to tipping are *<0.4* of the fraction of the population.**

.A number of models seem to show a bias toward very low critical mass sizes, which is not replicated in empirical studies and may suggest that dynamics or assumptions of these models are not realistic. They imply an overly-optimistic view of the potential for a critical mass to tip a system. It must be noted that in a large majority of models, the initial seed node, or first adopter of an alternative norm was normally taken to be 1 person, or a very small fraction *<5%* of the total population.




Fig. 7 demonstrates that the range in which tipping is most likely to occur is between 0 and 40% of the population ECDF (0.95) - empirical: 0.43, modelling: 0.52 respectively. This implies that values above this are either too close to a linear response to be considered tipping, or there is no critical mass at which the system tips for a given system state, i.e even at critical masses above this range there is no tipping possible.


As previously mentioned, a number of concepts find broad agreement in the literature, transcending disciplinary lines. Below in table x we synthesise some higher level takeaways in language which is more general and less technical.

| Key Concept | Findings | Implications | References |
| --- | --- | --- | --- |
| High-profile individuals (social influencers) | Influencers* may increase the possibility of a cascade under certain circumstances, but this effect is marginal, and can be polarising. Treatments specifically attempting to leverage these actors are often not cost or resource effective. Outcomes are also unpredictable. Moreover, these nodes can often hinder cascades as they are hard to tip due to lower payoffs or even penalties (politician, public figures etc.) | In order to maximise their efficiency, interventions or campaigns attempting to influence or effect behavioural change should not rely solely on highly-visible or renowned social actors. A random selection of actors may be more successful in contentious social changes. | (Watts and Dodds, 2007; Bakshy et al., 2011; Centola et al., 2018; Efferson et al., 2020; Bellotti et al., 2023; Hodas and Lerman, 2014; Watts, 2002) |
| Frequency of exposure | People require repeated exposures to an alternative norm in order to change. Despite the complexities which may surround the relationship between exposure and response, the number of exposures over a certain time | New or uncertain contexts, for example norms related to climate change or when the causal mechanism of norm change is unknown require a careful strategy. Any intervention should focus on repeated exposure, and ensuring that information about | All |



| | | | |
|---|---|---|---|
| | is by far the most robust predictor. . | the desired norm is reaching people. It should also focus on ensuring information is not lost in noise, i.e by avoiding overwhelmed channels such as social media. | |
| Trust | The strength of social connection heavily mediates the spread of contagion between individuals. This is not always the same as social proximity, but is often correlated with it. | Trusted information sources are more effective at changing norms in their social networks than untrusted sources. This relationship is more severe for controversial or important norm changes. When considering these issues, or intervention potentials, trusted individuals in relation to the target group should be identified and leveraged for change. | (Bellotti et al., 2023; Christakis and Fowler, 2008, 2007; Fowler and Christakis, 2008; Iacopini et al., 2022; Nishioka and Hasegawa, 2022; Okada et al., 2022; Watts and Dodds, 2007)O |
| Network structure | Structured networks are more conducive to social tipping under complex contagion. Although this varies with size, structural traits such as homophily and clustering allow a seed to amplify itself, or gather critical mass size in order to initiate a successful cascade. | Tight-night, trusting and close communities are necessary to allow a sufficient build up of momentum for social change. This becomes more important for more controversial norm changes or those which provide a lower personal reward, or even a penalty. | (Okada et al., 2022; Bellotti et al., 2023; Centola, 2013; Watts and Dodds, 2007) |
| Type and context of norm change | Contagion dynamics differ substantially depending on what type of behaviour or norm change is occurring, as well as the context. For | Different societal norm changes require different solutions. These relationships should be explicitly studied on a per norm basis. For example for consumption, flying, | (Bastos et al., 2013; Bellotti et al., 2023; Christakis and Fowler, 2008, 2007; Efferson et al., 2020; Fink et al., |





| | | | |
|---|---|---|---|
| | example educated people have a strong impact on others smoking behaviour, but village elders or knowledge-holders only have a marginal influence over health behaviour, where the household is most important. exogenous intervention. Group identity may reduce the effect of exogenous attempts at norm change e.g policy. Different people may be more or less responsive to change depending on these circumstances. | or driving behaviour. Policy interventions should rely on this knowledge. | 2015; Fowler and Christakis, 2008; Hodas and Lerman, 2014; Romero et al., 2011) |
| Personal Preferences and Heterogeneity | Similar to the above, personal preferences for a specific norm, and their distribution in a population can affect the success of a cascade in a population. This is not just limited to how strongly different fractions of a population feel towards a certain norm, it also relates to the distribution of these feelings. For example, increasing the variance of this distribution of preferences tends to reduce norm spread. | Understanding the distributions of preferences in terms of changing norms needs to be considered in mass-scale changes of social norms. This is most relevant for governance and the policy-maker. With this information, intervention strategies can target groups with preferences who are more likely to facilitate endogenous norm spread. | (Efferson et al., 2020; Fahimipour et al., 2022; Karsai et al., 2016; Wiedermann et al., 2020) |

**Table 5: Critical mechanisms affecting social tipping processes on networks as identified by their frequency in the literature.**



## 5.0 Discussion and Conclusion

Although complex contagion dynamics on networks are generally not amenable to reductionist methods of analysis (Shaliz & Thomas, 2011), our results show that there is broad agreement across the literature on particular variables affecting the success of contagion. Chiefly among these are clustering and structure in networks topology, and a high degree of trust between social connections (Fig. 4). These factors are also critical in instances where norm change is difficult, payoffs for switching norms are low, there is social pressure from the in-group, or the norm is connected to social identity (Efferson, 2020). All of these things are now more relevant than ever where our existing societal norms are no longer fit for the purpose of living in harmony with our planet and her boundaries (Otto et al., 2020b). A particularly relevant issue is the strong tie of group identity to problematic behavioural norms, which stymie the endogenous spread of social norms even after a targeted intervention (Efferson et al., 2020; Ehret et al., 2022). In the light of climate change these behavioural norms could correspond to things such as driving a large car, flying, or eating meat (Peattie, 2010a). Social tipping points research in SES calls for leveraging social tipping points for rapid societal change (Milkoreit, 2023; Winkelmann et al., 2020), but it doesn't go into whether tipping is even possible for some behavioural norms, or the dynamics which are required for particularly recalcitrant or sensitive behavioural norms. Our review shows clearly that each norm change is highly dependent on social context, individual preference distribution and heterogeneity. It also shows that a high variance in the distribution of personal preferences (social polarisation) is detrimental to changing social norms, which is an ever more pressing issue (Frei et al., 2023). Despite the complexities mentioned above, we see a clear non-linear trend when we investigate the critical mass required to induce tipping in a social network (Fig 6). More concretely, we display evidence that a critical mass of around 25% of the population can precipitate a population wide tipping event. The results here also serve to answer to some extent our research question by supporting the potential existence of a Pareto effect in social tipping dynamics. Although this can, and should not be generalised to all realms of social norms and social systems, it is a helpful indicator and target to aim for if policy makers are attempting to push, or monitor wide-scale social change. Perhaps an interesting case to watch is the rise of vegetarianism in Germany. Figures currently show the percent of the population who are vegetarian at around 10% (Statista, 2023), which is in the range for tipping as shown by Fig 7, has an increasing rate of change. A more generalizable result is given by Fig 7, which gives an estimate for the lower and upper ranges where tipping may occur. Additionally, the good agreement between empirical evidence and modelling results observed in this work supports the predictive power of models in their investigation of social contagion processes. This is particularly positive considering all modelling results shown in Fig 6 each used different types and forms of models. These ranged from equation based modelling, to system dynamics and agent based modelling with probabilistic approaches. Empirical validation of these modelling approaches is important for their inclusion in high level or integrated modelling frameworks, which generally require reliable heuristics which can be scaled up to an aggregate level, such as in IAMs (Trutnevyte et al., 2019). For modellers going a step beyond, and introducing more social complexity in large models (Donges et al., 2020), the validation across modelling approaches may guide approaches which are less computational intensive without losing accuracy. Further research would ideally seek to





replicate the existence of a pareto effect with a more rigorous statistical investigation considering a significantly larger
number of studies. Moreover, by identifying critical factors for successful social contagion across a broad scope of literature, this paper acts as a guide for where future research on topics should occur. For example where there is a lack of agreement in the discipline, as in Fig.6 regarding factors like network connectivity, population size, and/or global information.

A key issue affecting this analysis was the sample size, particularly with respect to the tipping points results in section 4. The
dimensionality, heterogeneity and scale of variables relating to complex contagion on social networks across disciplines is such that it becomes 1. prohibitively difficult to process, categorise, and harmonise findings across disciplines, and 2. difficult to say with certainty what factors contribute positively or negatively to enabling social tipping. In this sense our work should be considered as an agenda setting narrative-review, and by no means an exhaustive survey of the literature. We also only consider a one dimensional aspect of social tipping, namely its reliance on critical mass as a time dependent
variable. Additionally, we neglect multistability, and assume there is no intermittent or regressive behaviour of the system once it is tipped, which is a substantial issue to consider (Ferraz de Arruda et al, 2023). Although we attempted to cover most common network topologies, we decided that multi-layer networks were mostly out of the scope of this review due to the added complexity normally found in these approaches. Higher level network structures have a non-trivial effect on contagion dynamics (De Domenico, 2023; Zhang et al., 2023) and the field of social tipping and social contagion in general would
benefit from a comparison between these structures and typical, or single layer network structures. Finally, our investigation was necessarily bound to the domain of social networks, and we didn't consider work with alternative mapping between social entities, or those that neglect this entirely. Many of the reviewed models are not always integrated in broader SES systems themselves; either energy use, emissions, or environmental behaviour are absent. Future research should be directed at reconciling or refining this gap between conceptual frameworks and integrated modelling, where more generic tipping
dynamics are included in an SES model. Recent global SES models, or World Earth Models (WEMs) which explicitly simulate social dynamics on a micro scale (Donges et al., 2020), and contributions from ecological economics (Lamperti et al., 2018) are good starting points, and have laid the foundation for further inroads. Despite the complexities mentioned above, our macroscopic approach to measuring tipping thresholds provides concrete guidance in the form of a range of possibility for the critical mass required to facilitate social tipping events on social networks. Where causality is important,
we also supplement this more approximate range with a causal investigation of the factors contributing to social tipping. Our focus on complex contagion, and recalcitrant norm change mean our recommendations are well placed to aid the navigation of inherently difficult societal transitions like the one to net-zero. On the flipside, in situations where the norm change is minor and possible, our range of tipping thresholds provides a concrete, and empirically supported target for the policy maker trying to encourage the spread of easier to palate sustainable norm change in a social group.





**Code/Data availability**

All data and code used to run the analysis, produce the figures, and harmonise the datasets can be found on
https://github.com/foroveralls/pareto_tipping.

**Acknowledgements**

JFD is grateful for financial support by the European Research Council Advanced Grant project ERA (Earth Resilience in
the Anthropocene, ERC-2016-ADG-743080) and the European Union's Horizon research and innovation programme under
grant agreement No 101081661, project WorldTrans - Transparent Assessment for Real People.

The authors thank Wolfram Barfuss for helpful suggestions on literature to be included in this review.

**Author contributions**

J.E, I.M.O, and J.F.D. developed the conceptual framework. J.E performed the literature review, data analysis, developed the
figures, and led the writing of the manuscript with contributions from I.M.O, and input from J.F.D.

**Conflict of interest statement**

At least one of the (co-)authors is a member of the editorial board of Earth System Dynamics.

**Appendix A**

| Term | Explanation |
| --- | --- |
| Avg degree | The average number of connections per node in the network. |
| Clustering | The degree to which nodes in a network tend to cluster together. |
| Degree Heterogeneity | The variability in the number of connections that nodes in the network have. |
| Density | The proportion of actual connections to the number of possible connections within the network. |
| Geographical proximity | The closeness in geographical location between nodes in a network. |
| Global info/random rewiring | The availability of global information in the network and the formation of random connections. |



| Global information/weak ties | The role of weak ties in providing access to global information. |
|---|---|
| Homogenously mixing network | A network where nodes are equally likely to connect with each other. |
| Homophily | The tendency of individuals to associate and bond with similar others. |
| In-group conformity | The tendency of individuals to conform to the norms and behaviours of their respective groups. |
| Information transmission probability | The likelihood of information being successfully transmitted between connected nodes in the network. |
| Information transmission rate | The rate at which information is transmitted through the network. |
| Jointness of supply | The extent to which the supply of a good or service is shared among individuals. |
| Lattice | A structured network topology where each node is connected to its nearest neighbors. |
| Linkage probability | The probability of a connection forming between two nodes in the network. |
| Memory length | The amount of past information that nodes in the network retain. |
| Network size | The number of nodes in the network. |
| Node degree (out) | The number of outgoing connections from a node. |
| Node degree (in) | The number of incoming connections to a node. |
| Population size | The total number of individuals within a given population or network. |
| Public awareness | The level of knowledge and awareness among the public or nodes in the network. |
| Random rewiring | The process of randomly rearranging connections within the network. |
| Seed degree (out) | The number of outgoing connections from the initial or seed nodes. |
| Seed social influence | The level of influence exerted by the seed nodes. |
| Social influence of seed nodes | The extent to which seed nodes can influence the norms and behaviours of other nodes in the network. |
| Social proximity | The closeness of nodes in the network based on social relationships. |
| Social referents | Influential individuals or nodes within the network that serve as reference points for others. |
| Spreader influence | The ability of specific nodes, termed spreaders, to propagate information or norms efficiently within the network. |
| Structure | The arrangement of nodes and connections within the network. |
| Stubbornness | The resistance of nodes to change their state or adopt new norms and behaviours. |




| Threshold heterogeneity | The diversity in the thresholds that nodes have for adopting new norms or behaviours. |
|---|---|
| Tie strength | The intensity or closeness of the relationships between connected nodes. |
| Time window | The specific period considered for observing and analysing the dynamics of the network. |
| Trust | The level of confidence nodes share regarding the choice of their norms |
| Trust (in group) | The level of trust that individuals have within their respective groups or clusters in the network. |
| Weak ties | The connections between nodes that are not very strong or close. |
| Zealots | Highly committed or fervent nodes in the network that actively propagate, or resist the propagation of specific norms or beliefs. |


**Table A1: A glossary of terms relevant tom our literature review and analysis which may provide the reader with additional context to understand Fig. 5 and Fig. 6 in the main text.**

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
