# Peer review of "The Pareto effect in tipping social networks: from minority to majority"

_EGUsphere, 2023_

## Author Response (AR1)

**Anonymous Referee #1:**

Notes to editor/reviewers:
  ➢ Due to the number of changes made to the manuscript in terms of technical corrections (typos, grammar, sentence structure etc.), we have only left the changes which are relevant to the reviewer comments and the respective corrections in the markup.
  ➢ Figures 5 and 4 have been combined, this combined product is Figure 4.
  ➢ Figures 6 and 7 have been combined, this combined product is Figure 5.

Firstly, we would like to thank you for the numerous helpful comments and suggestions you have made on our manuscript. It was indeed challenging in places.

This is a very interesting work that combines a challenging literature review with some results. I say "challenging" because the topic (tipping in social networks) demands several expertises that can be quite far from each other. Network science and social network analysis, dynamical systems (phase transitions, non-linear dynamics, etc.), socio-ecological systems, simulation, data analysis, and so on.

The manuscript makes a good job at introducing the core concepts needed for such discussion, and the effort to integrate all those, coming from different disciplines, is generally successful: someone unfamiliar with social networks, information cascades, etc. will achieve a sufficient understanding of the research problem.

Beyond this general positive impression, the manuscript suffers from some shortcomings that should be addressed. In some sense, these weaknesses are "technical" (writing style, typos), but they end up affecting the message, and not only the form. For example, we can find some sentences which are unfinished or out of context (example: line 210: "Although in the former it is the spread of cooperation.". It makes reference to the previous text, but this sentence ill-formed: it reads as if something had to follow after 'cooperation'. Yet another example (line 260): "An example is reducing" (the sentence is unfinished, it simply ends there).

We completely agree that in general there are numerous "technical" weaknesses throughout the manuscript that require refinement. With full disclosure, we were under some time constraints due to the Global Tipping Points report due to be published and the manuscript polish did suffer. We apologize if these errors added difficulties in reading the manuscript and plan to rectify these systemically in an updated manuscript.  For example, the line in 210 will be removed and the lines in the previous sentence will be updated to the effect of "..the magnitude and rate of contagion(cooperation in the former) spread.". Sentences such as the example from line 260 will naturally be removed. There is also duplicates of some table entries e.g. in Table 3 which will be removed.

Completed:

➢ We have checked the entire manuscript for writing style and typos and made substantial edits (see markup).
➢ The specific examples in line 210, and 260 have been removed.
➢ Table 4 entry duplicates have been removed
➢ All tables have been corrected for formatting. writing style and typos.

I am aware that these are easy-to-solve issues. There are more (perhaps not so obvious). My point is that such formal errors interfere with the smoothness of the text --which needs to be really smooth if meant as a review for scholars from other disciplines. These errors (and many other smaller ones) suggest that the text was written with some hastiness.

Again, we agree with the comments and the importance of readability in a review article. Amongst addressing redundancies, repetitions and typographical errors, we plan to streamline and condense the amount of text in general, especially in section 2. Which addresses the first part of your comment below

A second indication of hastiness is the fact that some paragraphs are redundant (they can be some pages apart). I think that the concept of "threshold" is a good illustration of this: both their macro and individual dimensions keep appearing in different places, and so such a central concept is at risk of becoming "fuzzy" (except for the explicit and clear definitions in Box 1). My suggestion is then to reconsider the style and organization, specially in Sec. 2, to optimize the pedagogical value of the work.

If you are referring to examples such as (line 185) "An important distinction regarding thresholds is to be made between the system level macroscopic tipping threshold and individual agent thresholds, or threshold fractions.", where the "individual agent threshold" may confuse or add to this fuzziness, we acknowledge this and plan to remove potentially confusing or superfluous definitions.

Completed (refers to the previous two comments):
➢ Section two has been heavily edited (See markup), See lines 112-120. Lines 144-157 have been almost entirely re-written with a more explicit definition of social tipping. We have removed the definition of tipping using spillovers, as well as system dynamics, and focused on one approach. We have also removed references to criticality in the definition of tipping (Line 165) and simplified our justification of the use of the second derivative.
➢ We have simplified the language of thresholds. Threshold fraction has been renamed to "Individual Threshold" as it is more intuitive. The other definitions have been renamed (see Box 1) to improve understanding. We have also improved the consistency of the use of individual and macroscopic thresholds in the text to reduce "fuzziness".

I recommend caution as well when using certain words. In Box 1, we see "percolation" and "spreading" as equivalent (verbatim: "[...] which after a percolation (spreading) process must

occupy [...]". Although percolation is behind the study of contagion-like processes on networks (disease, information), it has a wider meaning in network science and statistical physics.

We agree, in Watts (2002), he refers to percolation models of (disease) spreading as belonging to "a larger class of contagion problems". Based on this interpretation, percolation would not have equivalence with spreading but rather be a subclass of a contagion process. The 'spreading' was added as an aid to interdisciplinary or non-specialist readers. We will either add a footnote in the box which explains the term and its context or remove it and add information on percolation in another section of 2.1.

Completed:
  ➢ The implication of equivalence (spreading) has been removed from the text.

For the sake of transparency and reproducibility, details about data collection (Sec. 3.1) should be more precise. The literature search must have for sure retirned hundreds, if not thousands, of titles. How then the authors reached an initial number of 33?

There are certainly a few errors in this section. For example, in line 374 we say that the literature considered for the first 3 stages is N = 41, when it should in fact be N = 39. The data collection section is also missing information which has led to the confusion you mention. Firstly, we only looked at the first 4 pages of google scholar results. Secondly the search string ""network" *, "networks"* AND ("complex contagion" OR "norm diffusion")," is left over from an earlier draft and this search was not used. Thirdly, we only selected articles which, along with the other existing criteria, only referred to social contagion between people. These things will be amended and should provide a better template to reproduce the literature search used in the manuscript. We also plan to include a figure detailing the filtering process of literature in each stage 1, 2, and 3.

Completed:
  ➢ Older search terms have been removed (lines 396-398)
  ➢ Other criteria have been added at lines 398 i.e the focus on contagion in human networks.

To sum up, the value of a document like this lies in its clarity. Precisely because the scientific content of the manuscript is very appealing, the effort to communicate it must be equated.

We wholeheartedly agree. As mentioned in these replies, we will systemically update the manuscript to remove technical errors such as those mentioned above and focus on clarity of concepts. Key attention will be given to reducing "fuzziness" in terms or notions and removing redundant or multiple definitions where possible.

See "completed" sections above. Additionally, we have removed several potentially distracting passages including superfluous terms, notions or definitions. Examples are:

- The removal of an additional definition of tipping point in section 2.2 (see lines 252-260 under All Markup.)
- Clarification of how social tipping and network theoretic terms relate to each other, and usage of definitions from network theory (Lines 259-263, All Markup).
- Clarification of terminology regarding tipping points in the methods (Lines 489-495, All Markup), as well as Box 1.
- Additional clarification of the use of social tipping in existing literature and how it relates to our usage and analysis (Lines 113-120, All Markup).
- Provided additional definitions where they were missing to help understand, e.g. definition of "wide bridges" added in 2.2 (Lines 246-253, All Markup).
- Additional clarification of terms in section 2.2 (See All Markup).

Finally, other minor issues to take into account:

- The "Clustered lattice" representation in Fig. 2 is very counterintuitive. I am sure a better representation is possible.

This will be updated to a standard representation where the lattice is in a circular formation
Completed: Lattice has been redrawn (See Fig. 2) to be more in line with conventional depictions.

- Table 1: Erdös-Rényi networks have a low average path length.

This will be corrected.
Completed.

- Figure 3: please add labels to x- and y-axis

They are indeed missing and will be added.
Completed.

- Please revise the references format. One can find, e.g., "PNAS", "Proceedings of the National Academy of Sciences"; missing publication years; etc.

We will go through the references to ensure they are correct.
Completed: Abbreviations have been updated to be in line with the Caltech Library guidelines as stipulated within ESD guidelines.

References
Watts, D. J.: A simple model of global cascades on random networks, Proc. Natl. Acad. Sci., 99, 5766–5771, https://doi.org/10.1073/pnas.082090499, 2002.

**Referee #2: Sibel Eker**

Notes to editor/reviewers:
- ➤ Due to the number of changes made to the manuscript in terms of technical corrections (typos, grammar, sentence structure etc.), we have only left the changes which are relevant to the reviewer comments and the respective corrections in the markup.
- ➤ Figures 5 and 4 have been combined, this combined product is Figure 4.
- ➤ Figures 6 and 7 have been combined, this combined product is Figure 5.

This manuscript presents a literature review on social norm diffusion/contagion on social networks, and whether a critical mass similar to a Pareto effect is observed. The manuscript makes a relevant and important contribution to the expanding literature on social tipping by aggregating the available empirical evidence on tipping dynamics. I also appreciate that authors clarify and describe tipping-related concepts and terms in Section 2, yet I believe that manuscript would benefit from a more condensed version of it.

My main concern is how "social tipping" is implicitly defined in the manuscript, and that it is not clearly aligned with the real-life examples of what social tipping in the climate change and sustainability context entails. For instance, the manuscript focuses on social networks and complex contagion, but does it mean that all social tipping process, such as rapid reduction of EV battery costs, coverage of climate change in the school curricula, as identified in the authors' own paper from 2020, have to include contagion dynamics through a social network? Even if we can rightfully represent many of those in networks, do we have to? Is the presence of a network structure a necessary condition for tipping? How does this relate to the existing social tipping and "positive" tipping literature that does not necessarily focus on networks? Answering such questions would improve the manuscript.

Firstly a big thanks for the considerate and in-depth comments and feedback. They will certainly help us improve the manuscript.

Regarding the focus on social tipping, the questions you pose relate to the underlying difficulty in disentangling complex systems. Quickly summed, as mentioned in lines starting from 170, we seek to use the tools of network theory as a basis to discuss and investigate social tipping processes. Inasmuch as social interaction is required for social tipping, where the former implies some network structure, we don't have to represent it as such. We do this because of the large body of evidence supporting it as a useful tool for doing so (Berner et al., 2023; Guilbeault et al., 2018; Sayama et al., 2013; Smaldino, 2023, Winkelmann et al., 2022). We hope to relate to existing literature by seeing how previous findings, and social tipping in general can also be explained through network theory, which may improve or provide alternate understandings vs existing methods. Also to clarify; this paper focuses purely on the social element of tipping, i.e. regarding social change. We do not consider socio-economic, technological, or other kinds of tipping.

I hope that helps answer your questions.

We will also include these answers in the text at your suggestion, as we agree these are not explained enough.

Completed:
➢ Added a paragraph similar to our above response in section 1 (lines 68-85) addressing answers to all questions listed by the reviewer.

**Relatively minor comments:**

Lines 75-88 : This paragraph does a fair job in listing the objectives, yet they sound a bit too ambitious. For instance, "we identify critical factors influencing tipping processes in social systems" is eventually only limited to a handful of studies included in your review on social networks, therefore I suggest reformulating these objectives more consistently with the actual methods and results.

We agree and will amend these statements to be restricted to what our methods and results show.

Completed:
➢ We have changed this section to more accurately reflect our results*. For example, in the sentence from the reviewer comment we change "social systems" to "subsystems" and critical factors to "several critical factors".

*Some edits are not included in markup due to technical problems with MS word version history.

Figure 1a:  I don't find this figure very clear, especially considering the accompanying text. If the "shaded area represents the domain where the system exhibits a nonlinear response" would any nonlinear curve above the y=x line mean "tipping"? How about the zone below the line? It might also include very nonlinear changes.

We agree this is not really communicated properly. Firstly any curve above or below the y=x line would be a "spillover" or some form of non-linear response but it's ambiguous as to whether this means tipping per our definitions. We will address this. Secondly, the zone below the line also represents the opposite of a "spillover", also a non linear change. We will address this by defining the line above y=x as a positive non-linear change, or a "positive spillover", and below the line a negative non-linear change, or "negative spillover" we will address whether this constitutes tipping in the sense of our own definitions and refer more specifically to how this graph relates. We will also address the text in line 115 to 125 which does not make this clear.

Completed:
➢ Figure 1a has been completely re-envisioned along with our more explicit definitions of social tipping added into section 2.1. We have included a strict delineation on the figure to concur with our new definition of social tipping as per lines 120-142 and Box 1. The accompanying text has been rewritten to more clearly assist in understanding the figure.

Line 121-122: "where roughly a 20% change in a system control parameter results in an 80% change in the system state at equilibrium". I am not sure if this is the right

formulation of your idea. What happens as lambda moves from 0.1 to 0.12, which is a 20% change in the control parameter?

This is correct and the statement is indeed misleading and wrong. The section you point to has a few issues we already plan to address, such as the unconventional use of the current adopter fraction as a system parameter and not a variable, as well as the potentially confusing use of a cdf. We will also amend the lines to indicate the change is referring not to the current system control parameter but to the maximum potential fraction of adopters.

Completed: We have removed these lines.

Line 126-127: "the shown Pareto CDF depicts a scenario in which a minority of actors have convinced a large majority to switch to another social norm." I think it doesn't show that one, because the y axis is F when t goes to infinity, the final state. When lambda=0.8, we end up at >0.8 and no tipping occurs. The pareto cdf line covers many different scenarios not necessarily those where a minority convinces the others.

We agree that Figure 1.a is currently a bit ambiguous and potentially misleading. As stated above, we will address these concerns as part of the whole section, which includes a slight redesign of the plot for a few reasons including that stated above.
The axis points (0,0) are also mismatched and will be corrected. We are currently considering two potential directions for the plot. Both use more traditional approaches similar to recursion or cobweb plots from system dynamics. We refer to two examples from the literature: the former, shown in Figure 2, and the latter, shown in Figure 1, of the respective papers(Efferson et al., 2020; Wiedermann et al., 2020). Hopefully using existing and familiar representations will aid understanding, reduce confusion, and clarify our main message with the plot.

Completed:
  ➤ See response to Figure 1a comment above. We have replaced the Pareto CDF with a more general non-linear function. We specifically highlight the scenario we are intending to show with the blue "Social tipping zone". It is now clear that when the function goes through this area it is fulfilling our requirements for social tipping.
  ➤ We have also removed the language of spillovers in reference to our tipping point concept (lines 150-165, All Markup), and thus this plot to reduce any misunderstanding or confusion for the reader.

Figure 1b: Appreciating this plot of tipping dynamics over time, yet it is the typical s-curve of transitions/innovation adoption. Could you expand on what makes it "tipping"? Furthermore, it is not clear on the figure what the tipping threshold is, as suggested in Box 1.

The point of this figure is to try and reconcile the concepts that are introduced or will be introduced in the paper with concepts existing in the literature on social tipping. Similar to the transition/innovation and adoption, social tipping events resolve as the typical s-curve due to inherently similar spreading dynamics which occur in these systems. Not to mention that the former usually occur through social interaction.

What this figure represents under the "tipping phase" can be seen as tipping under the notion of the original authors from where we draw the language for each phase. You are correct if you are implying it does not qualify for our definition of "Tipping event" as per Box 1, where the figure 1b shows the transient dynamics, and our definition is based on the steady state. This is an oversight and will be addressed so as to reduce ambiguity. One approach would be to state that social tipping used in the sense of the notions from the original authors has a slightly different meaning to our definition. Another would be to provide a different definition of the "social tipping" stage, for example more closely drawing on the definitions(section 4.2 and SI) from work by Winkelamnn, et al. (2022).

Completed:
  ➤ We have expanded section 2.1 to include some characteristics of tipping (not only definitions) (110-119) and the text surrounding the figure (lines 135-205.) to more strongly demonstrate how this figure is demonstrating social tipping.
  ➤ Added a vertical line notated "Tipping point t_c" to indicate where the system demonstrates the beginning of tipping as per our definitions in Box 1. Note: the definitions in Box 1 have slightly changed and this notation reflects the new definitions.

Line 182: Using cascade and social tipping interchangeably. "Cascade" has been used (in the GTPR) recently to refer to the tipping dynamics in different systems affecting each other, both for climate tipping points and positive tipping points. This terminology choice might be confusing for the readers

We agree and note that we plan to improve the clarity and strictness in the use of terms in general to reduce confusion. Regarding this specific point, we will either remove the concurrent use of cascade or change the term to something less
confusing such as "contagion".

Completed:
  ➤ While we appreciate the potential point for confusion, we have addressed this by specifically noting that the use of the term "cascade" is from network theory in the the first sections (2.1, 2.2) of section 2. We have also added clarification in lines 260-264 (All Markup), and in lines 240-243 (All Markup).

Figure 3: Not clear what the x- and y- axes represent.

The y axis represents the probability of the population having a certain threshold fraction which is represented by the x axis. These labels are missing and will be added.

Completed: Missing x and y axis labels added (see Figure 3).

Line 368: 21 papers were discarded because they did not include complex contagion. It would be useful to mention what they included instead.

This is an oversight and will be explicitly stated, but we imply here that they include only simple contagion dynamics.

Completed:
  ➢ Lines added to 403 referring to the above.

Table 4: Very useful table summarizing the reviewed studies. To strengthen the conclusions of the manuscript, we need to know more about these studies, though. Could you specify sample size/network size, geographic location, what the nodes and edges are, socioeconomic and demographic background etc.?

We agree, however a lot of these things are left out due to the economy of space on what is already a very full manuscript. We will address this by adding an extended version of the table with important characteristics in a supplementary file. Additionally we will possibly add more columns to Table 4. according to their perceived importance in the literature.

Completed:
  ➢ We have changed the format of the table (portrait to landscape) to fit more information.
  ➢ We have included network/sample size, as well as methodology. The latter is also related to another comment below. The nodes are always people as stated in the methodology. The edge type was unfortunately not recorded specifically but is assumed to be some form of social tie. Demographic background can be inferred from the bracketed information in the Network Topology column, otherwise the reader is referred to the source. We state this in line 427.
  ➢ A full table with all of our gathered information will be provided in the SI.

Figure 6: Very informative figure especially because it distinguishes between model-based and empirical studies! The sample size is confusing, though, since the caption says that it includes 87 papers and the text earlier mentioned the eventual # of studies in the review were ~30. Please clarify. It raises the question of what made the empirical studies with a similar lambda around 20% but not tipped (bottom left). This figure could have been improved in a bubble chat format, where the dot size refers to the network size in each study. "Population size" came up as one of the most important drivers in Fig.4, and we do observe its relevance. For instance, the global percentage of the vegetarian population is 20%, but no tipping is observed.

Firstly thank you for the useful suggestion regarding the dot size representing the network size, we will implement this in the figures.The number 87 in Fig 6. Refers to the number of trajectories from the studies in the last stage of the review which were reduced to 12 articles. As you indicate, this is not distinguished properly and will be made clear through specific notation i.e N_p for papers and N_t for trajectories. The small group of empirical studies that did not tip, and in fact ended up with less supporters than at the critical mass illustrates how tipping is highly dependent on the social system and context. We will include an entry in the results which addresses this cluster and explains the factors that may have prevented a successful positive tipping event.

Completed (refers now to Fig. 5a):

> ➤ We have changed the notation for the final fraction of adopters from n/N to F_inf to reduce confusion between trajectories and sample size (number of papers). The figure and description have been updated to reflect this.
> ➤ We have added text to address the small group of non-tipped data points in lines 507-510.
> ➤ The population size was not able to be incorporated into the figure due to data processing issues. However it has now been included in Table 4 as an extra column for the reader to cross-reference with.

Line 518- Evidence of critical mass: For the empirical studies covered in this paper, it would be very informative for the strength of evidence whether they are in lab settings or in a real-life experimental setting or contain large-scale data?

We agree and this will be added in some form, either in another column of Table 4, an additional table, or a Table entry in the appendix.

Completed:
> ➤ Refer to Table 4 comment. We have added the setting (lab, online, real-life), along with a more detailed methodology classification, i.e modelling, observational, empirical.

Line 531-533 "For modellers…" : Could you expand on "validation across modelling approaches" for computational efficiency? There are examples of system dynamics models which do not use a complex contagion and threshold approach, but show that the inflection point of the norm adoption function (the counterpart of the cdf of the probability of an agent adopting a norm for the fraction of agents who already adopted it) is the most important driver of large scale change in the diet context. How do we for instance cross-validate those? https://www.nature.com/articles/s41893-019-0331-1

As touched on in previous responses, a complex contagion approach is a modeling decision to represent a type of phenomena in this case social contagion or tipping processes. System dynamics models as you mentioned are also a modelling decision to do the same. In this case where the underlying processes are the same, i.e norm adoption, implying social interaction, it is often possible to directly compare representations of dynamics. In the case you mentioned, the norm adoption function is equivalent to the CDF as you already identified. It is in theory possible to construct this CDF even from a distribution of microscopic thresholds (threshold fractions) to obtain one function which governs the adoption rate in a social group undergoing a contagion event (Wiedermann et al., 2020). This would allow one to compare these two things across models. I believe this is beyond the scope of our paper but we will amend this section with more detail on how cross validation can be performed across model types under certain assumptions.

Completed:
> ➤ We have addresses this in a more comprehensive version of our above response in lines 573-579, explicitly referring to the sigmoid adoption curve and existing methods to compare and validate these curves across scientific disciplines and sub-fields.

Line 542: Undermining what you did.

This is indeed perhaps underselling the results and will be reformulated to be less severe or removed.

Completed: Line has been removed.

**Minor comments:**

Line 45 "mechanisms. In which…" please watch the grammar.

This will be amended.
Completed.

Line 174 Please clarify what "systems-dynamics" is, since it is not a commonly used term.

This will be changed to system dynamics.
Completed.

Line 338: Which "Ref"?

This reference is missing and will be added in.
Completed.

I suggest to put table captions above the table, since it is a more commonly used convention.

We will adopt this convention.
Completed.

Line 497: table x

This will be changed to Table 5

Completed.

The first paragraph of Section 5: I suggest to divide this paragraph and several others, since they are too lengthy and contain more than 1 argument, main point etc.

We agree, section 5 and in general throughout the paper will be condensed and more structured, i.e broken up into multiple paragraphs.

Completed: Section 5 has been broken up into multiple paragraphs.

**References**

Berner, R., Gross, T., Kuehn, C., Kurths, J., and Yanchuk, S.: Adaptive Dynamical Networks, https://doi.org/10.48550/arXiv.2304.05652, 12 April 2023.

Efferson, C., Vogt, S., and Fehr, E.: The promise and the peril of using social influence to reverse harmful traditions, Nat. Hum. Behav., 4, 55–68, https://doi.org/10.1038/s41562-019-0768-2, 2020.

Guilbeault, D., Becker, J., and Centola, D.: Complex Contagions: A Decade in Review, in: Complex Spreading Phenomena in Social Systems: Influence and Contagion in Real-World Social Networks, edited by: Lehmann, S. and Ahn, Y.-Y., Springer International Publishing, Cham, 3–25, https://doi.org/10.1007/978-3-319-77332-2_1, 2018.

Sayama, H., Pestov, I., Schmidt, J., Bush, B. J., Wong, C., Yamanoi, J., and Gross, T.: Modeling complex systems with adaptive networks, Comput. Math. Appl., 65, 1645–1664, https://doi.org/10.1016/j.camwa.2012.12.005, 2013.

Smaldino, P.: Modeling Social Behavior: Mathematical and Agent-Based Models of Social Dynamics and Cultural Evolution, Princeton University Press, 360 pp., 2023.

Wiedermann, M., Smith, E. K., Heitzig, J., and Donges, J. F.: A network-based microfoundation of Granovetter's threshold model for social tipping, Sci. Rep., 10, 11202, https://doi.org/10.1038/s41598-020-67102-6, 2020.

Winkelmann, R., Donges, J. F., Smith, E. K., Milkoreit, M., Eder, C., Heitzig, J., Katsanidou, A., Wiedermann, M., Wunderling, N., and Lenton, T. M.: Social tipping processes towards climate action: A conceptual framework, Ecol. Econ., 192, 107242, https://doi.org/10.1016/j.ecolecon.2021.107242, 2022

---

## Author Response (AR2)

**Author response: revision Round 2 – started 04 Aug 2024 Anonymous Referee #3, Report #2:**

**General comments to editors:**

➢ We have restructured and modified the abstract for conciseness, and more specifically stated our findings relating to practical strategies to create norm change (lines 24-27) the content is essentially the same.

➢ There are some small technical corrections throughout the document, the content is the same.

1. In their analysis, the authors explore heterogeneity in individual thresholds, and Figure 3 provides possible threshold distributions. Recent work by Janas et al. (2024) elicits clear examples of such threshold distributions and heterogeneity in the real world. Their study could offer empirical evidence that complements the theoretical examples.

➢ We add a threshold distribution from Janas et al. to Figure 3, seen in panel (d). This provides the empirical perspective mentioned above.

➢ We have changed the text to incorporate this addition (lines 298-319). We discuss the implications of the distribution for social tipping and what it represents, along with some general technical improvements to the writing.

2. The paper would benefit from a more explicit discussion of future research opportunities. While the analysis is thorough, outlining potential avenues for further investigation would provide guidance for future work in this area. I particularly liked the time scales discussion and think future experiments should tackle the role of time in social tipping.

➢ The discussion and conclusion have been restructured to more explicitly discuss future research opportunities. Lines 594-630 have been heavily re-written with the previously scattered suggestions for future research removed from other sections and now consolidated and presented together.

➢ We created a new section specifically addressing temporality of social tipping processes (see lines 620-630), under which we include a discussion of time scales and relevant existing methodologies which are well suited to further research on the topic.

3. The emphasis on a 20-25% critical mass as a general tipping point could be misleading. There are many reasons why a common or typical range of critical mass may not exist across different contexts. This presents another opportunity for future research: testing the conditions under which tipping occurs at much lower or higher percentages would offer valuable insights.

➢ We have modified text in the conclusion (565-569) to highlight the limitations of our findings in terms of broader application and validity in various social systems. Specifically stating that critical mass estimates (20-25%) cannot be generalized to all scenarios.

➢ We have also modified the text in the abstract to a similar extent in lines 19-24. We use more explicit language, emphasizing that the tipping point of around 25% is found within our dataset, not "within susceptible social systems", which may be misleading. We have removed the word "general" in line 19 to be more specific. Lines were added (23-25) to indicate that our results show the "possibility" of rapid social change in certain contexts. This should indicate that this is not inevitable or assured in all contexts.